# A machine learning Automated Recommendation Tool for synthetic biology

Tijana Radivojević [1,2,3], Zak Costello[1,2,3], Kenneth Workman [1,3,4] & Hector Garcia Martin [1,2,3,5✉]

Synthetic biology allows us to bioengineer cells to synthesize novel valuable molecules such as renewable biofuels or anticancer drugs. However, traditional synthetic biology approaches involve ad-hoc engineering practices, which lead to long development times. Here, we present the Automated Recommendation Tool (ART), a tool that leverages machine learning and probabilistic modeling techniques to guide synthetic biology in a systematic fashion, without the need for a full mechanistic understanding of the biological system. Using sampling-based optimization, ART provides a set of recommended strains to be built in the next engineering cycle, alongside probabilistic predictions of their production levels. We demonstrate the capabilities of ART on simulated data sets, as well as experimental data from real metabolic engineering projects producing renewable biofuels, hoppy flavored beer without hops, fatty acids, and tryptophan. Finally, we discuss the limitations of this approach, and the practical consequences of the underlying assumptions failing.

[1] DOE Agile BioFoundry, Emeryville, CA 94608, USA. [2] Biofuels and Bioproducts Division, DOE Joint BioEnergy Institute, Emeryville, CA 94608, USA. [3] Biological Systems and Engineering Division, Lawrence Berkeley National Laboratory, Berkeley, CA 94720, USA. [4] Department of Bioengineering, University of California, Berkeley, CA 94720, USA. [5] BCAM, Basque Center for Applied Mathematics, Bilbao 48009, Spain. ✉email: hgmartin@lbl.gov

Metabolic engineering[1] enables us to bioengineer cells to synthesize novel valuable molecules such as renewable biofuels[2,3], or anticancer drugs[4]. The prospects of metabolic engineering to have a positive impact in society are on the rise, as it was considered one of the "Top Ten Emerging Technologies" by the World Economic Forum in 2016[5]. Furthermore, an incoming industrialized biology is expected to improve most human activities: from creating renewable bioproducts and materials, to improving crops and enabling new biomedical applications[6].

However, the practice of metabolic engineering has been far from systematic, which has significantly hindered its overall impact[7]. Metabolic engineering has remained a collection of useful demonstrations rather than a systematic practice based on generalizable methods. This limitation has resulted in very long development times: for example, it took 150 person-years of effort to produce the antimalarial precursor artemisinin by Amyris; and 575 person-years of effort for Dupont to generate propanediol[8], which is the base for their commercially available Sorona fabric[9].

Synthetic biology[10] aims to improve genetic and metabolic engineering by applying systematic engineering principles to achieve a previously specified goal. Synthetic biology encompasses, and goes beyond, metabolic engineering: it also involves non-metabolic tasks such as gene drives able to extinguish malaria-bearing mosquitoes[11] or engineering microbiomes to replace fertilizers[12]. This discipline is enjoying an exponential growth, as it heavily benefits from the byproducts of the genomic revolution: high-throughput multi-omics phenotyping[13,14], accelerating DNA sequencing[15] and synthesis capabilities[16], and CRISPR-enabled genetic editing[17]. This exponential growth is reflected in the private investment in the field, which has totalled ~$12B in the 2009–2018 period and is rapidly accelerating (~$2B in 2017 to ~$4B in 2018)[18].

One of the synthetic biology engineering principles used to improve metabolic engineering is the Design-Build-Test-Learn (DBTL[19,20]) cycle—a loop used recursively to obtain a design that satisfies the desired specifications (e.g., a particular titer, rate, yield or product). The DBTL cycle's first step is to design (D) a biological system expected to meet the desired outcome. That design is built (B) in the next phase from DNA parts into an appropriate microbial chassis using synthetic biology tools. The next phase involves testing (T) whether the built biological system indeed works as desired in the original design, via a variety of assays: e.g., measurement of production or/and omics (transcriptomics, proteomics, metabolomics) data profiling. It is extremely rare that the first design behaves as desired, and further attempts are typically needed to meet the desired specification. The Learn (L) step leverages the data previously generated to inform the next Design step so as to converge to the desired specification faster than through a random search process.

The Learn phase of the DBTL cycle has traditionally been the most weakly supported and developed[20], despite its critical importance to accelerate the full cycle. The reasons are multiple, although their relative importance is not entirely clear. Arguably, the main drivers of the lack of emphasis on the L phase are: the lack of predictive power for biological systems behavior[21], the reproducibility problems plaguing biological experiments[3,22–24], and the traditionally moderate emphasis on mathematical training for synthetic biologists.

Machine learning (ML) arises as an effective tool to predict biological system behavior and empower the Learn phase, enabled by emerging high-throughput phenotyping technologies[25]. By learning the underlying regularities in experimental data, machine learning can provide predictions without a detailed mechanistic understanding. Training data are used to statistically link an input (i.e., features or independent variables) to an output (i.e., response or dependent variables) through models that are expressive enough to represent almost any relationship. After this training, the models can be used to predict the outputs for inputs that the model has never seen before. Machine learning has been used to, e.g., predict the use of addictive substances and political views from Facebook profiles[26], automate language translation[27], predict pathway dynamics[28], optimize pathways through translational control[29], diagnose skin cancer[30], detect tumors in breast tissues[31], predict DNA and RNA protein-binding sequences[32], and drug side effects[33]. However, the practice of machine learning requires statistical and mathematical expertise that is scarce, and highly competed for ref. [34].

Here, we provide a tool that leverages machine learning for synthetic biology's purposes: the Automated Recommendation Tool (ART, Fig. 1). ART combines the widely used and general-purpose open-source scikit-learn library[35] with a Bayesian[36] ensemble approach, in a manner that adapts to the particular needs of synthetic biology projects: e.g., a low number of training instances, recursive DBTL cycles, and the need for uncertainty quantification. The data sets collected in the synthetic biology field (<100 instances) are typically not large enough to allow for the use of deep learning, so this technique is not currently used in ART. However, once high-throughput data generation[14,37] and automated data collection[38] capabilities are widely used, we expect data sets of thousands, tens of thousands, and even more instances to be customarily available, enabling deep learning capabilities that can also leverage ART's Bayesian approach. In general, ART provides machine learning capabilities in an easy-to-use and intuitive manner, and is able to guide synthetic biology efforts in an effective way.

The efficacy of ART in guiding synthetic biology is showcased through five different examples: one test case using simulated data, three cases where we leveraged previously collected experimental data from real metabolic engineering projects, and a final case where ART is used to guide a bioengineering effort to improve productivity. In the synthetic case and the three experimental cases where previous data are leveraged, we mapped one type of –omics data (targeted proteomics in particular) to production. In the case of using ART to guide experiments, we mapped promoter combinations to production. In all cases, the underlying assumption is that the input (–omics data or promoter combinations) is predictive of the response (final production), and that we have enough control over the system so as to produce any new recommended input. The test case permits us to explore how the algorithm performs when applied to systems that present different levels of difficulty when being "learnt", as well as the effectiveness of using several DTBL cycles. The real metabolic engineering cases involve data sets from published metabolic engineering projects: renewable biofuel production[39], yeast bioengineering to recreate the flavor of hops in beer[40], and fatty alcohols synthesis[41]. These projects illustrate what to expect under different typical metabolic engineering situations: high/low coupling of the heterologous pathway to host metabolism, complex/simple pathways, high/low number of conditions, high/low difficulty in learning pathway behavior. Finally, the fifth case uses ART in combination with genome-scale models to improve tryptophan productivity in yeast by 106% from the base strain, and is published in parallel[42] as the experimental counterpart to this article. We find that ART's ensemble approach can successfully guide the bioengineering process even in the absence of quantitatively accurate predictions (see e.g., the "Improving the production of renewable biofuel" section). Furthermore, ART's ability to quantify uncertainty is crucial to gauge the reliability of predictions and effectively guide recommendations towards the least known part of the phase space. These experimental metabolic engineering cases also illustrate how applicable the underlying assumptions are, and what happens when they fail.

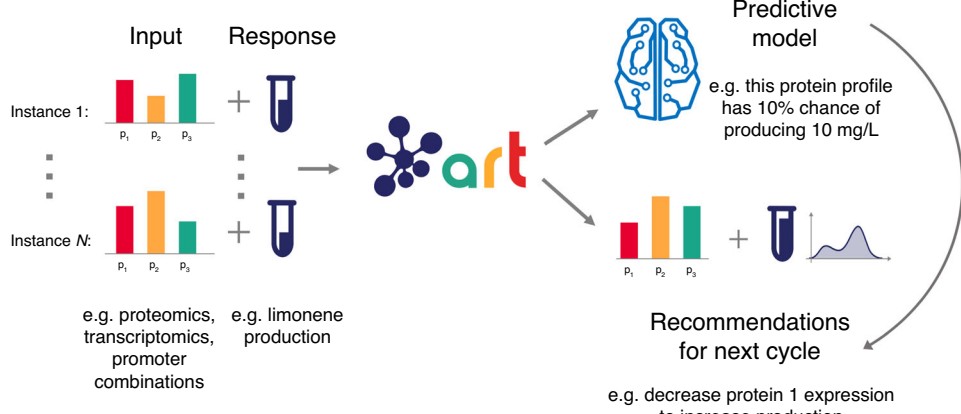

**Fig. 1 ART provides predictions and recommendations for the next cycle.** ART uses experimental data (input and responses in the left side) to (i) build a probabilistic predictive model that predicts response (e.g., production) from input variables (e.g., proteomics), and (ii) uses this model to provide a set of recommended inputs for the next experiment (new input) that will help reach the desired goal (e.g., increase response/production). The input phase space, in this case, is composed of all the possible combinations of protein expression levels (or transcription levels, promoters,… for other cases). The predicted response for the recommended inputs is characterized as a full probability distribution, effectively quantifying prediction uncertainty. Instances refer to each of the different examples of input and response used to train the algorithm (e.g., each of the different strains and/or conditions, that lead to different production levels because of different proteomics profiles). See Fig. 2 for details on the predictive model and Fig. 3 for details on the recommendation strategy. An example of the output can be found in Supplementary Fig. 5.

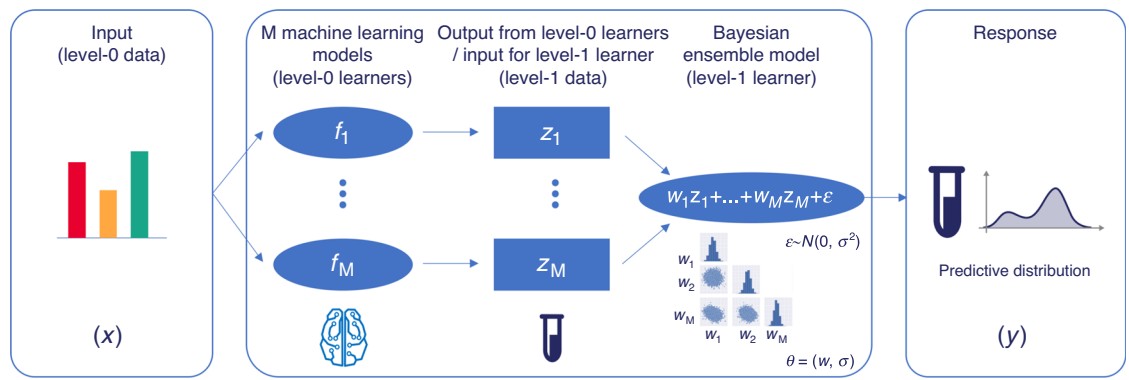

**Fig. 2 ART provides a probabilistic predictive model of the response.** ART combines several machine learning models from the scikit-learn library with a Bayesian approach to predict the probability distribution of the output. The input to ART is proteomics data (or any other input data in vector format: transcriptomics, gene copy, etc.), which we call level-0 data. This level-0 data is used as input for a variety of machine learning models from the scikit-learn library (level-0 learners) that produce a prediction of production for each model ($z_i$). These predictions (level-1 data) are used as input for the Bayesian ensemble model (level-1 learner), which weights these predictions differently depending on its ability to predict the training data. The weights $w_i$ and the variance $\sigma$ are characterized through probability distributions, giving rise to a final prediction in the form of a full probability distribution of response levels. This probabilistic model is the "Predictive model" depicted in Fig. 1.

In sum, ART provides a tool specifically tailored to the synthetic biologist's needs in order to leverage the power of machine learning to enable predictable biology. This combination of synthetic biology with machine learning and automation has the potential to revolutionize bioengineering[25,43,44] by enabling effective inverse design (i.e., the capability to design DNA to meet a specified phenotype: a biofuel productivity rate, for example). We have made a special effort to write this paper to be accessible to both the machine learning and synthetic biology readership, with the intention of providing a much-needed bridge between these two very different collectives. Hence, we have emphasized explaining basic machine learning and synthetic biology concepts, since they might be of use to a part of the readership.

## Results

**Key capabilities.** ART leverages machine learning to improve the efficacy of bioengineering microbial strains for the production of desired bioproducts (Fig. 1). ART gets trained on available data (including all data from previous DBTL cycles) to produce a model (Fig. 2) capable of predicting the response variable (e.g., production of the jet fuel limonene) from the input data (e.g., proteomics data, or any other type of data that can be expressed as a vector). Furthermore, ART uses this model to recommend new inputs (e.g., proteomics profiles, Fig. 3) that are predicted to reach our desired goal (e.g., improve production). As such, ART bridges the Learn and Design phases of a DBTL cycle.

ART can import data directly from Experimental Data Depo[45], an online tool where experimental data and metadata are stored in a standardized manner. Alternatively, ART can import EDD-style .csv files, which use the nomenclature and structure of EDD exported files (see the "Importing a study" section in Supplementary Information).

By training on the provided data set, ART builds a predictive model for the response as a function of the input variables

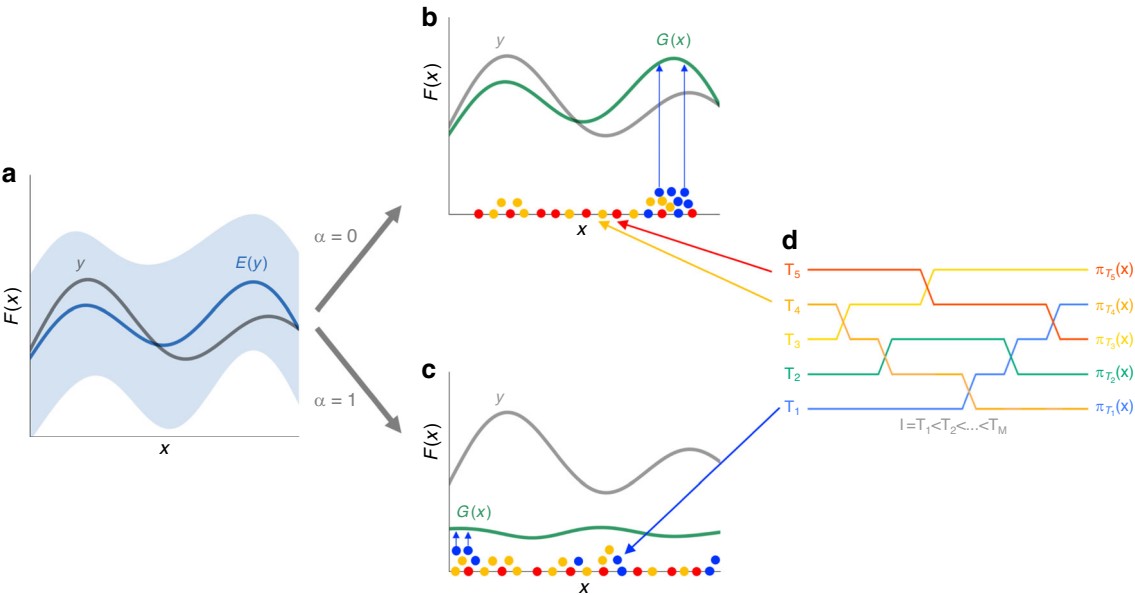

**Fig. 3 ART chooses recommendations by sampling the modes of a surrogate function.** The true response $y$ (e.g., biofuel production to be optimized) is shown as a function of the input $\mathbf{x}$ (e.g., proteomics data), as well as the expected response $E(y)$ after several DBTL cycles (**a**), and its 95% confidence interval (blue). Depending on whether we prefer to explore (**c**) the phase space where the model is least accurate or exploit (**b**) the predictive model to obtain the highest possible predicted responses, we will seek to optimize a surrogate function $G(\mathbf{x})$ (Eq. (5)), where the exploitation-exploration parameter is $\alpha = 0$ (pure exploitation), $\alpha = 1$ (pure exploration) or anything in between. Parallel-tempering-based MCMC sampling (**d**) produces sets of vectors $\mathbf{x}$ (colored dots) for different "temperatures": higher temperatures (red) explore the full phase space, while lower temperature chains (blue) concentrate in the nodes (optima) of $G(\mathbf{x})$. The exchange between different "temperatures" provides more efficient sampling without getting trapped in local optima. Final recommendations (upward-pointing blue arrows) to improve response are provided from the lowest temperature chain, and chosen such that they are not too close to each other and to experimental data (at least 20% difference). These recommendations are the "Recommendations for next cycle" depicted in Fig. 1. In this example, they represent protein expression levels that should be targeted to achieve predicted production levels. See Fig. 7 for an example of recommended protein profiles and their experimental tests.

(Fig. 2). Rather than predicting point estimates of the response variable, ART provides the full probability distribution of the predictions (i.e., the distribution for all possible outcomes for the response variable and their associated probability values). This rigorous quantification of uncertainty enables a principled way to test hypothetical scenarios in-silico, and to guide the design of experiments in the next DBTL cycle. The Bayesian framework chosen to provide the uncertainty quantification is particularly tailored to the type of problems most often encountered in metabolic engineering: sparse data which is expensive and time-consuming to generate.

With a predictive model at hand, ART can provide a set of recommendations expected to produce the desired outcome, as well as probabilistic predictions of the associated response (Fig. 3). ART supports the following typical metabolic engineering objectives: maximization of the production of a target molecule (e.g., to increase Titer, Rate, and Yield, TRY), its minimization (e.g., to decrease the toxicity), as well as specification objectives (e.g., to reach a specific level of a target molecule for a desired beer taste profile[40]). Furthermore, ART leverages the probabilistic model to estimate the probability that at least one of the provided recommendations is successful (e.g., it improves the best production obtained so far), and derives how many strain constructions would be required for a reasonable chance to achieve the desired goal.

Although ART can be applied to the problems with multiple output variables of interest, it currently supports only the same type of objective for all output variables. Hence, it does not yet support the maximization of one target molecule along with the minimization of another (see "Success probability calculation" in Supplementary Information).

**Using simulated data to test ART**. Synthetic data sets allow us to test how ART performs when confronted by problems of different difficulty and dimensionality, as well as gauge the effectiveness of the experimental design of the initial training set and the availability of training data. In this case, we tested the performance of ART for 1–10 DBTL cycles, three problems of increasing difficulty ($F_E$, $F_M$, and $F_D$, see Fig. 4), and three different dimensions of input space ($D = 2$, 10, and 50, Fig. 5). We simulated the DBTL processes by starting with a training set given by 16 instances (Fig. 1), obtained from Fig. 4 functions. Different instances, in this case, may represent different engineered strains or different induction or fermentation conditions for a particular strain. The choice of initial training set is very important (Supplementary Fig. 3).

The initial input values were chosen as Latin Hypercube[46] draws, which involves dividing the range of variables in each dimension into equally probable intervals and then choosing samples such that there is only one in each hyperplane (hyper-row/hyper-column) defined by those intervals. This ensures that the set of samples is representative of the variability of the input phase space. A less careful choice of initial training data can significantly hinder learning and production improvement (Supplementary Fig. 3). In this regard, a list of factors to consider when designing an experiment can be found in the "Designing optimal experiments for machine learning" section in Supplementary Information. We limited ourselves to the maximization case and, at each DBTL cycle, generated 16 recommendations that maximized the objective function given by Eq. (5). This choice mimicked triplicate experiments in the 48 wells of throughput of a typical automated fermentation platform[47]. We employed a tempering strategy for the exploitation-exploration

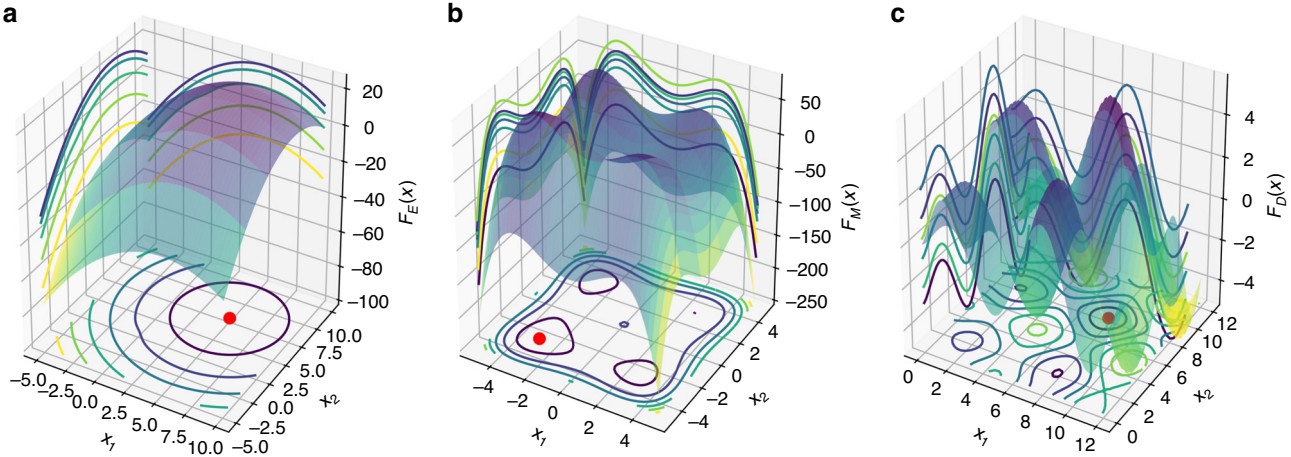

**Fig. 4 Synthetic data test functions for ART.** These functions present different levels of difficulty to being "learnt", and are used to produce synthetic data and test ART's performance (Fig. 5). **a** $F_E(\mathbf{x}) = -\frac{1}{d}\sum_i^d (x_i - 5)^2 + \exp\left(-\sum_i x_i^2\right) + 25$; **b** $F_M(\mathbf{x}) = \frac{1}{d}\sum_i^d (x_i^4 - 16x_i^2 + 5x_i)$; **c** $F_D(\mathbf{x}) = \sum_i^d \sqrt{x_i}\sin(x_i)$.

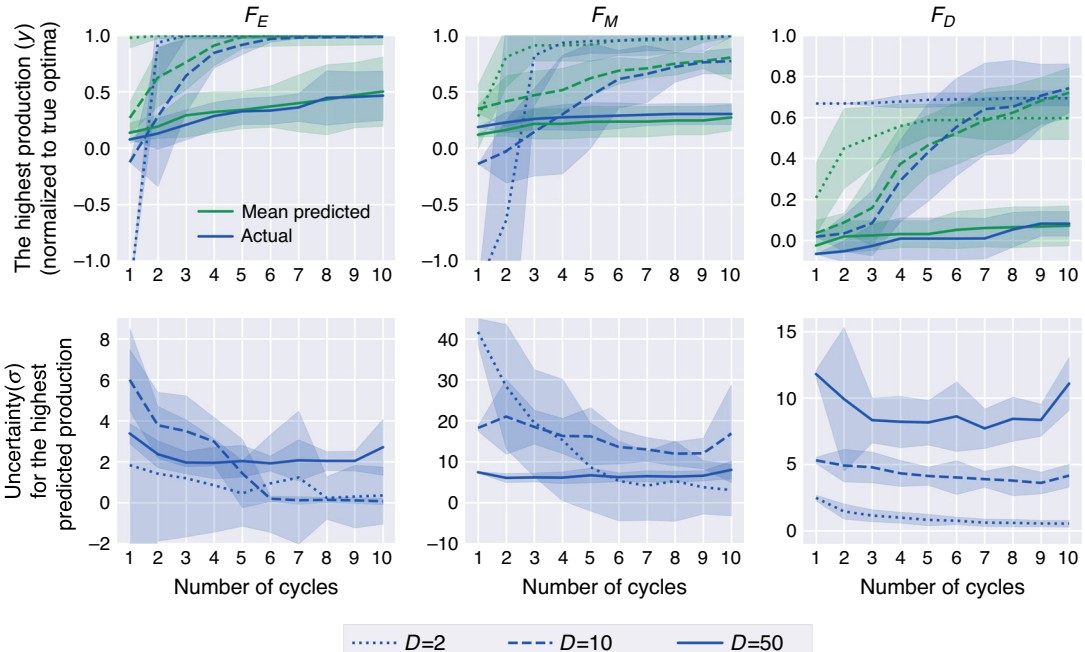

**Fig. 5 ART performance improves significantly beyond the usual two DBTL cycles.** Here we show the results of testing ART's performance with synthetic data obtained from functions of different levels of complexity (Fig. 4), different phase space dimensions (2, 10, and 50), and different amounts of training data (DBTL cycles). The top row presents the results of the simulated metabolic engineering in terms of highest production achieved so far for each cycle (as well as the corresponding ART predictions). The production increases monotonically with a rate that decreases as the problem is harder to learn, and the dimensionality increases. The bottom row shows the uncertainty in ART's production prediction, given by the standard deviation of the response distribution (Eq. (2)). This uncertainty decreases markedly with the number of DBTL cycles, except for the highest number of dimensions. In each plot, lines and shaded areas represent the estimated mean values and 95% confidence intervals, respectively, over ten repeated runs. Mean Absolute Error (MAE) and training and test set definitions can be found in Supplementary Fig. 4.

parameter, i.e., assigned $\alpha = 0.9$ at the start for an exploratory optimization, and gradually decreased the value to $\alpha = 0$ in the final DBTL cycle for the exploitative maximization of the production levels.

ART performance improves significantly as more data are accrued with additional DTBL cycles. Whereas the prediction error, given in terms of mean average error (MAE), remains constantly low for the training set (i.e., ART is always able to reliably predict data it has already seen), the MAE for the test data (data ART has not seen) in general decreases markedly with the addition of more DBTL cycles (Supplementary Fig. 4). The

exceptions are the most complicated problems: those exhibiting highest dimensionality ($D = 50$), where MAE stays approximately constant, and the difficult function $F_D$, which exhibits a slower decrease. Furthermore, the best production among the 16 recommendations obtained in the simulated process increases monotonically with more DBTL cycles: faster for easier problems and lower dimensions and more slowly for harder problems and higher dimensions. Finally, the uncertainty in those predictions decreases as more DBTL cycles proceed (Fig. 5). Hence, more data (DBTL cycles) almost always translates into better predictions and production. However, we see that these benefits are

rarely reaped with only the two DBTL cycles customarily used in metabolic engineering (see examples in the next sections): ART (and ML in general) becomes only truly efficient when using 5–10 DBTL cycles.

Different experimental problems involve different levels of difficulty when being learnt (i.e., being predicted accurately), and this can only be assessed empirically. Low dimensional problems can be easily learnt, whereas exploring and learning a 50-dimensional landscape is very slow (Fig. 5). Difficult problems (i.e., less monotonic landscapes) take more data to learn and traverse than easier ones. We will showcase this point in terms of real experimental data when comparing the biofuel project (easy) versus the dodecanol project (hard) below. However, there is no systematic way to decide a priori whether a given problem will be easy or hard to learn—the only way to determine this is by checking the improvements in prediction accuracy as more data is added. In any case, a starting point of at least ~100 instances is highly recommendable to obtain proper statistics.

**Improving the production of renewable biofuel**. The optimization of the production of renewable biofuel limonene through synthetic biology will be our first demonstration of ART using real-life experimental data. Renewable biofuels are almost carbon neutral because they only release into the atmosphere the carbon dioxide that was taken up in growing the plant biomass they are produced from. Biofuels are seen as the most viable option for decarbonizing sectors that are challenging to electrify, such as heavy-duty freight and aviation[48].

Limonene is a molecule that can be chemically converted to several pharmaceutical and commodity chemicals[49]. If hydrogenated, it displays characteristics that are ideal for next-generation jet-biofuels and fuel additives that enhance cold weather performance[50,51]. Limonene has been traditionally obtained from plant biomass, as a byproduct of orange juice production, but fluctuations in availability, scale, and cost limit its use as biofuel[52]. The insertion of the plant genes responsible for the synthesis of limonene in a host organism (e.g., a bacteria), however, offers a scalable and cheaper alternative through synthetic biology. Limonene has been produced in *E. coli* through an expansion of the celebrated mevalonate pathway (Fig. 1a in ref. [53]), used to produce the antimalarial precursor artemisinin[54] and the biofuel farnesene[55], and which forms the technological base on which the company Amyris was founded (valued ~$300M ca. 2019). This version of the mevalonate pathway is composed of seven genes obtained from such different organisms as *S. cerevisiae*, *S. aureus*, and *E. coli*, to which two genes have been added: a geranyl-diphosphate synthase and a limonene synthase obtained from the plants *A. grandis* and *M. spicata*, respectively.

For this demonstration, we use historical data from ref. [39], where 27 different variants of the pathway (using different promoters, induction times and induction strengths) were built. Data collected for each variant involved limonene production and protein expression for each of the nine proteins involved in the synthetic pathway. These data were used to feed Principal Component Analysis of Proteomics (PCAP)[39], an algorithm using the principal component analysis to suggest new pathway designs. The PCAP recommendations used to engineer new strains, resulting in a 40% increase in production for limonene, and 200% for bisabolene (a molecule obtained from the same base pathway). This small amount of available instances (27) to train the algorithms is typical of synthetic biology/metabolic engineering projects. Although we expect automation to change the picture in the future[25], the lack of large amounts of data has determined our machine learning approach in ART (i.e., no deep neural networks).

ART is able to not only recapitulate the successful predictions obtained by PCAP improving limonene production, but also improves significantly on this method. Firstly, ART provides a quantitative prediction of the expected production in all of the input phase space, rather than qualitative recommendations. Secondly, ART provides a systematic method that is automated, requiring no human intervention to provide recommendations. Thirdly, ART provides uncertainty quantification for the predictions, which PCAP does not. In this case, the training data for ART are the concentrations for each of the nine proteins in the heterologous pathway (input), and the production of limonene (response). The objective is to maximize limonene production. We have data for two DBTL cycles, and we use ART to explore what would have happened if we have used ART instead of PCAP for this project.

We used the data from DBLT cycle 1 to train ART and recommend new strain designs (i.e., protein profiles for the pathway genes, Fig. 6). The model trained with the initial 27 instances provided reasonable cross-validated predictions for the production of this set ($R^2 = 0.44$), as well as the three strains which were created for DBTL cycle 2 at the behest of PCAP (Fig. 6). This suggests that ART would have easily recapitulated the PCAP results. Indeed, the ART recommendations are very close to the PCAP recommendations (Fig. 7). Interestingly, we see that while the quantitative predictions of each of the individual models were not very accurate, they all signaled towards the right direction in order to improve production, showing the importance of the ensemble approach (Fig. 7). Hence, we see that ART can successfully guide the bioengineering process even in the absence of quantitatively accurate predictions.

Training ART with experimental results from DBTL cycles 1 and 2 results in even better predictions ($R^2 = 0.61$), highlighting the importance of the availability of large amounts of data to train ML models. This new model suggests new sets of strains predicted to produce even higher amounts of limonene. Importantly, the uncertainty in predicted production levels is significantly reduced with the additional data points from cycle 2.

**Brewing hoppy beer without hops by bioengineering yeast**. Our second example involves bioengineering yeast (*S. cerevisiae*) to produce hoppy beer without the need for hops[40]. To this end, the ethanol-producing yeast used to brew the beer, was modified to also synthesize the metabolites linalool (L) and geraniol (G), which impart hoppy flavor (Fig. 2 in ref. [40]). Synthesizing linalool and geraniol through synthetic biology is economically advantageous because growing hops is water and energetically intensive, and their taste is highly variable from crop to crop. Indeed, a startup (Berkeley Brewing Science, https://www.crunchbase.com/organization/berkeley-brewing-science#section-overview) was generated from this technology.

ART is able to efficiently provide the proteins-to-production mapping that required three different types of mathematical models in the original publication, paving the way for a systematic approach to beer flavor design. The challenge is different in this case as compared to the previous example (limonene): instead of trying to maximize production, the goal is to reach a particular level of linalool and geraniol so as to match a known beer tasting profile (e.g., Pale Ale, Torpedo or Hop Hunter, Fig. 8). ART can provide this type of recommendations, as well. For this case, the inputs are the expression levels for the four different proteins involved in the pathway, and the response is the concentrations of the two target molecules (L and G), for which we have desired targets. We have data for two DBTL cycles involving 50 different strains/instances (19 instances for the first DBTL cycle and 31 for the second one, Fig. 8). As in the previous

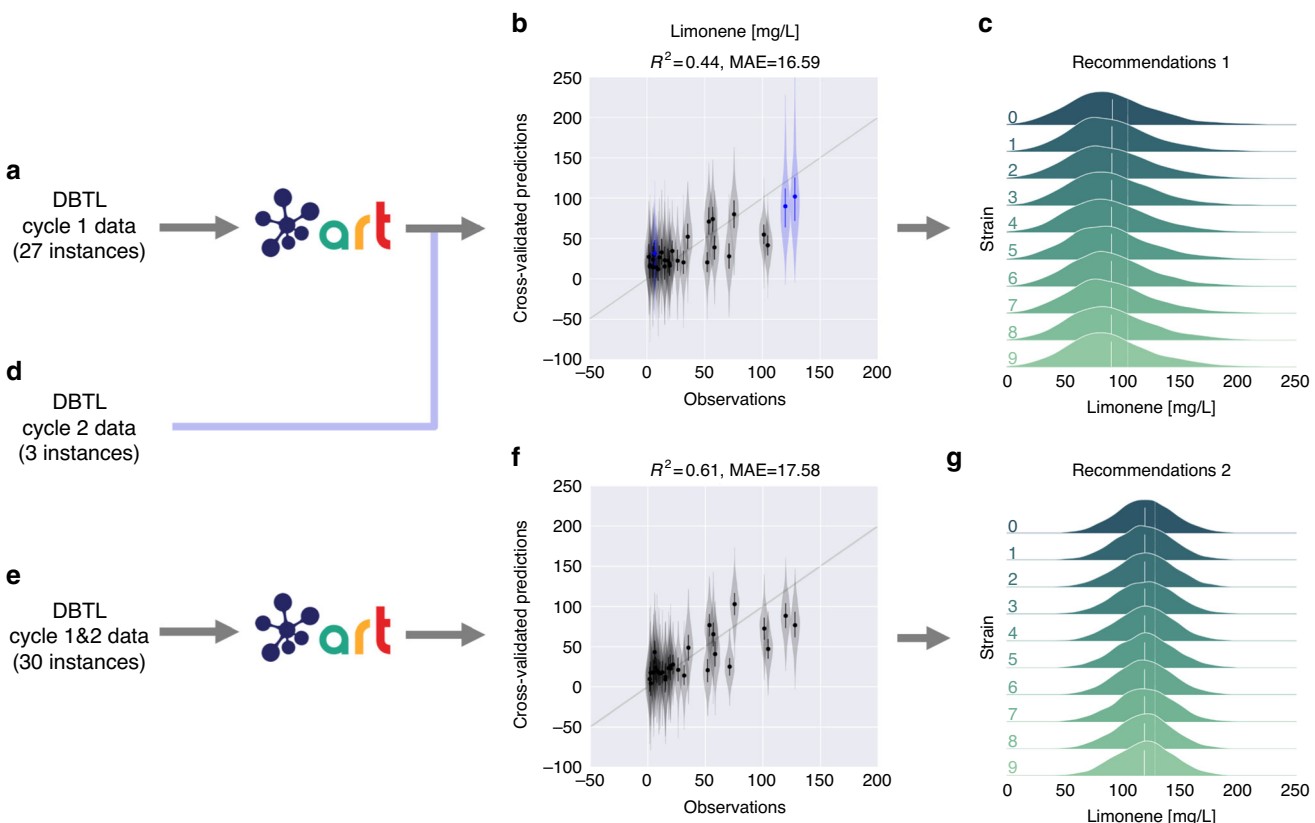

**Fig. 6 ART provides effective recommendations to improve biofuel production.** We used the first DBTL cycle data (**a**) to train ART and recommend new protein targets with predicted production (**c**). The ART recommendations were very similar to the protein profiles that eventually led to a 40% increase in production (Fig. 7). ART predicts mean production levels for the second DBTL cycle strains (**d**), which are very close to the experimentally measured values (three blue points in **b**). Adding those three points from DBTL cycle 2 provides a total of 30 strains for training (**e**) that lead to recommendations predicted to exhibit higher production and narrower distributions (**g**). Uncertainty for predictions is shown as probability distributions for recommendations (**c**, **g**) and violin plots for the cross-validated predictions (**b**, **f**). The cross-validation graphs (present in Figs. 8, 9 and Supplementary Figs. 8, 9, too) represent an effective way of visualizing prediction accuracy for data the algorithm has not yet seen. The closer the points are to the diagonal line (predictions matching observations) the more accurate the model. The training data are randomly subsampled into partitions, each of which is used to validate the model trained with the rest of the data. The black points and the violins represent the mean values and the uncertainty in predictions, respectively. $R^2$ and mean absolute error (MAE) values are only for cross-validated mean predictions (black data points).

case, we use this data to simulate the outcomes we would have obtained in case ART had been available for this project.

The first DBTL cycle provides a very limited number of 19 instances to train ART, which performs passably on this training set, and poorly on the test set provided by the 31 instances from DBTL cycle 2 (Fig. 8). Despite this small amount of training data, the model trained in DBTL cycle 1 is able to recommend new protein profiles that are predicted to reach the Pale Ale target (Fig. 8). Similarly, this DBTL cycle 1 model was almost able to reach (in predictions) the L and G levels for the Torpedo beer, which will be finally achieved in DBTL cycle 2 recommendations, once more training data is available. For the Hop Hunter beer, recommendations from this model were not close to the target.

The model for the second DBTL cycle leverages the full 50 instances from cycles 1 and 2 for training and is able to provide recommendations predicted to attain two out of three targets. The Pale Ale target L and G levels were already predicted to be matched in the first cycle; the new recommendations are able to maintain this beer profile. The Torpedo target was almost achieved in the first cycle, and is predicted to be reached in the second cycle recommendations. Finally, Hop Hunter target L and G levels are very different from the other beers and cycle 1 results, so neither cycle 1 or 2 recommendations can predict protein inputs achieving this taste profile. ART has only seen two

instances of high levels of L and G and cannot extrapolate well into that part of the metabolic phase space. ART's exploration mode, however, can suggest experiments to explore this space.

Quantifying the prediction uncertainty is of fundamental importance to gauge the reliability of the recommendations, and the full process through several DBTL cycles. In the end, the fact that ART was able to recommend protein profiles predicted to match the Pale Ale and Torpedo taste profiles only indicate that the optimization step (see the "Optimization-suggesting next steps" section) works well. The actual recommendations, however, are only as good as the predictive model. In this regard, the predictions for L and G levels shown in Fig. 8 (right side) may seem deceptively accurate, since they are only showing the average predicted production. Examining the full probability distribution provided by ART shows a very broad spread for the L and G predictions (much broader for L than G, Supplementary Fig. 6). These broad spreads indicate that the model still has not converged and that recommendations will probably change significantly with new data. Indeed, the protein profile recommendations for the Pale Ale changed markedly from DBTL cycle 1–2, although the average metabolite predictions did not (left panel of Supplementary Fig. 7). All in all, these considerations indicate that quantifying the uncertainty of the predictions is important to foresee the smoothness of the optimization process.

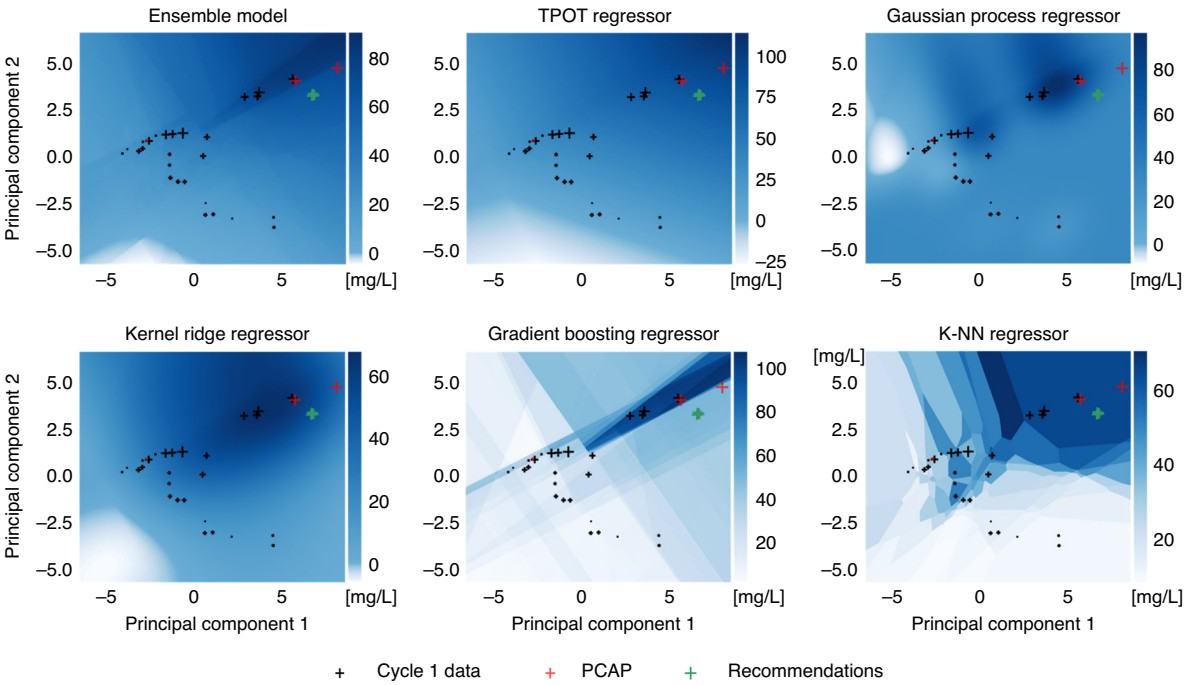

**Fig. 7 All algorithms point similarly to improve limonene production, despite quantitative differences.** Cross sizes indicate experimentally measured limonene production in the proteomics phase space (first two principal components shown from principal component analysis, PCA). The color heatmap indicates the limonene production predicted by a set of base regressors and the final ensemble model (top left) that leverages all the models and conforms the base algorithm used by ART. Although the models differ significantly in the actual quantitative predictions of production, the same qualitative trends can be seen in all models (i.e., explore upper right quadrant for higher production), justifying the ensemble approach used by ART. The ART recommendations (green) are very close to the protein profiles from the PCAP paper[39] (red), which were experimentally tested to improve production by 40%. Hence, we see that ART can successfully guide the bioengineering process even in the absence of quantitatively accurate predictions.

At any rate, despite the limited predictive power afforded by the cycle 1 data, ART recommendations guide metabolic engineering effectively. For both of the Pale Ale and Torpedo cases, ART recommends exploring parts of the proteomics phase space that surround the final protein profiles, which were deemed close enough to the desired targets in the original publication. Recommendations from cycle 1 and initial data (green and red in Supplementary Fig. 7) surround the final protein profiles obtained in cycle 2 (orange in Supplementary Fig. 7). Finding the final protein target becomes, then, an interpolation problem, which is much easier to solve than an extrapolation one. These recommendations improve as ART becomes more accurate with more DBTL cycles.

**Improving dodecanol production.** The final example is one of failure (or at least a mitigated success), from which as much can be learnt as from the previous successes. Ref. [41] used machine learning to drive two DBTL cycles to improve the production of 1-dodecanol in *E. coli*, a medium-chain fatty acid used in detergents, emulsifiers, lubricants, and cosmetics. This example illustrates the case in which the assumptions underlying this metabolic engineering and modeling approach (mapping proteomics data to production) fail. Although a ~20% production increase was achieved, the machine learning algorithms were not able to produce accurate predictions with the low amount of data available for training, and the tools available to reach the desired target protein levels were not accurate enough.

This project consisted of two DBTL cycles comprising 33 and 21 strains, respectively, for three alternative pathway designs (Fig. 1 in ref. [41], Supplementary Table 4). The use of replicates increased the number of instances available for training to 116 and 69 for cycles 1 and 2, respectively. The goal was to modulate the protein expression by choosing Ribosome Binding Sites (RBSs, the mRNA sites to which ribosomes bind in order to translate proteins) of different strengths for each of the three pathways. The idea was for the machine learning to operate on a small number of variables (~3 RBSs) that, at the same time, provided significant control over the pathway. As in previous cases, we will show how ART could have been used in this project. The input for ART, in this case, consists of the concentrations for each of three proteins (different for each of the three pathways), and the goal was to maximize 1-dodecanol production.

The first challenge involved the limited predictive power of machine learning for this case. This limitation is shown by ART's completely compromised prediction accuracy (Fig. 9). We hypothesize the causes to be twofold: a small training set and a strong connection of the pathway to the rest of host metabolism. The initial 33 strains (116 instances) were divided into three different designs (50, 31, and 35 instances, Supplementary Table 4), decimating the predictive power of ART (Fig. 9 and Supplementary Figs. 8 and 9). Now, it is complicated to estimate the number of strains needed for accurate predictions because that depends on the complexity of the problem to be learnt (see the "Using simulated data to test ART" section). In this case, the problem is harder to learn than the previous two examples: the mevalonate pathway used in those examples is fully exogenous (i.e., built from external genetic parts) to the final yeast host and hence, free of the metabolic regulation that is certainly present for the dodecanol producing pathway. The dodecanol pathway depends on fatty acid biosynthesis which is vital for cell survival (it produces the cell membrane), and has to be therefore tightly regulated[56]. We hypothesize that this characteristic makes it more difficult to learn its behavior by ART using only dodecanol

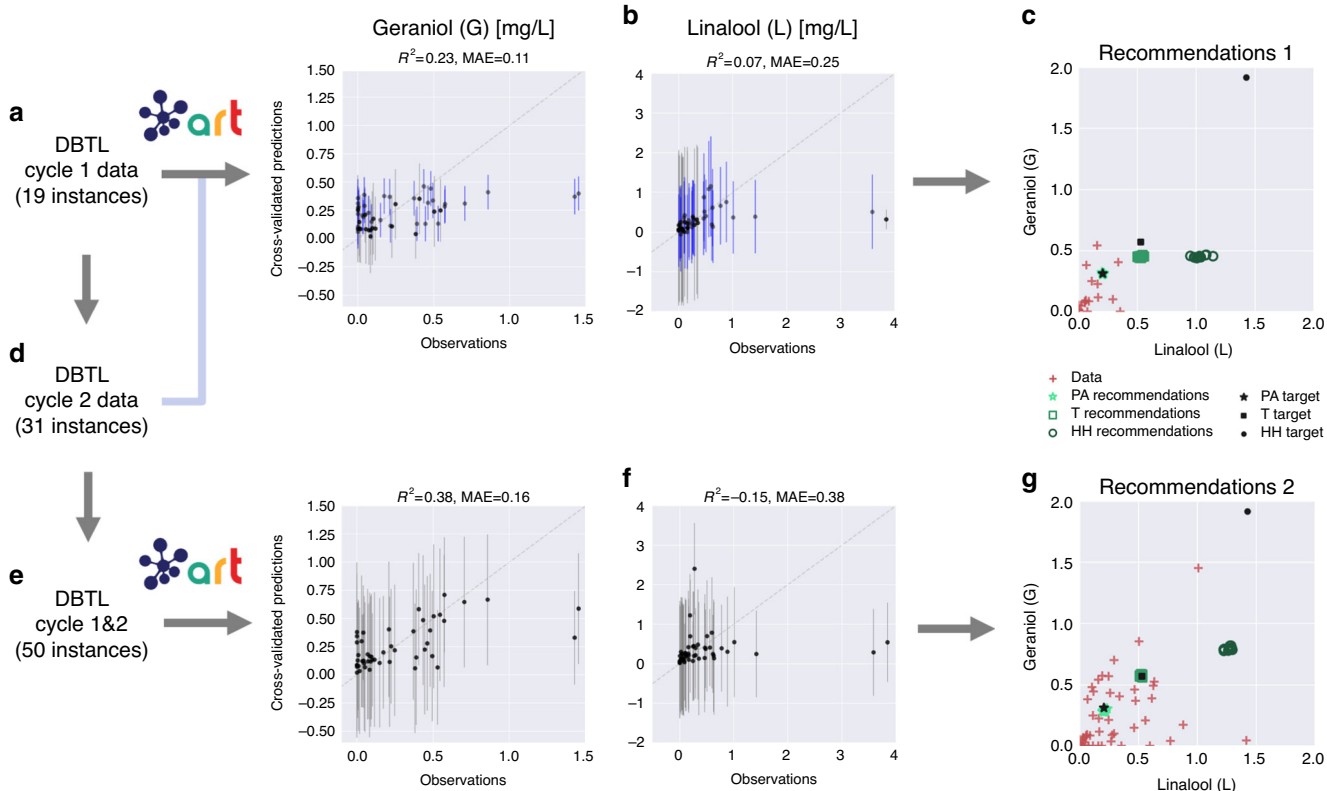

**Fig. 8 ART produces effective recommendations to bioengineer yeast to produce hoppy beer.** The 19 instances in the first DBTL cycle (**a**) were used to train ART, but it did not show an impressive predictive power (particularly for L (**b**)). In spite of it, ART is still able to recommend protein profiles predicted to reach the Pale Ale (PA) target flavor profile, and others that were close to the Torpedo (T) metabolite profile (**c** green points showing mean predictions). Adding the 31 strains for the second DBTL cycle (**d**, **e**) improves predictions for G but not for L (**f**). The expanded range of values for G & L provided by cycle 2 allows ART to recommend profiles which are predicted to reach targets for both beers (**g**), but not Hop Hunter (HH). Hop Hunter displays a very different metabolite profile from the other beers, well beyond the range of experimentally explored values of G and L, making it impossible for ART to extrapolate that far. Notice that none of the experimental data (red crosses) matched exactly the desired targets (black symbols), but the closest ones were considered acceptable. $R^2$ and mean absolute error (MAE) values are for cross-validated mean predictions (black data points) only. Bars indicate 95% credible interval of the predictive posterior distribution.

synthesis pathway protein levels (instead of adding also proteins from other parts of host metabolism).

A second challenge, compounding the first one, involves the inability to reach the target protein levels recommended by ART to increase production. This difficulty precludes not only bioengineering, but also testing the validity of the ART model. For this project, both the mechanistic (RBS calculator[57,58]) and machine learning-based (EMOPEC[59]) tools proved to be very inaccurate for bioengineering purposes: e.g., a prescribed sixfold increase in protein expression could only be matched with a twofold increase. Moreover, non-target effects (i.e., changing the RBS for a gene significantly affects protein expression for other genes in the pathway) were abundant, further adding to the difficulty. Although unrelated directly to ART performance, these effects highlight the importance of having enough control over ART's input (proteins in this case) to obtain satisfactory bioengineering results.

A third, unexpected, challenge was the inability of constructing several strains in the Build phase due to toxic effects engendered by the proposed protein profiles (Supplementary Table 4). This phenomenon materialized through mutations in the final plasmid in the production strain or no colonies after the transformation. The prediction of these effects in the Build phase represents an important target for future ML efforts, in which tools like ART can have an important role. A better understanding of this phenomenon may not only enhance bioengineering but also reveal new fundamental biological knowledge.

These challenges highlight the importance of carefully considering the full experimental design before leveraging machine learning to guide metabolic engineering.

## Discussion

ART is a tool that not only provides synthetic biologists easy access to machine learning techniques, but can also systematically guide bioengineering and quantify uncertainty. ART takes as input a set of training instances, which consist of a set of vectors of measurements (e.g., a set of proteomics measurements for several proteins, or transcripts for several genes) along with their corresponding systems responses (e.g., associated biofuel production) and provides a predictive model, as well as recommendations for the next round (e.g., new proteomics targets predicted to improve production in the next round, Fig. 1).

ART combines the methods from the scikit-learn library with a Bayesian ensemble approach and MCMC sampling, and is optimized for the conditions encountered in metabolic engineering: small sample sizes, recursive DBTL cycles and the need for uncertainty quantification. ART's approach involves an ensemble where the weight of each model is considered a random variable with a probability distribution inferred from the available data. Unlike other approaches, this method does not require the ensemble models to be probabilistic in nature, hence allowing us to fully exploit the popular scikit-learn library to increase

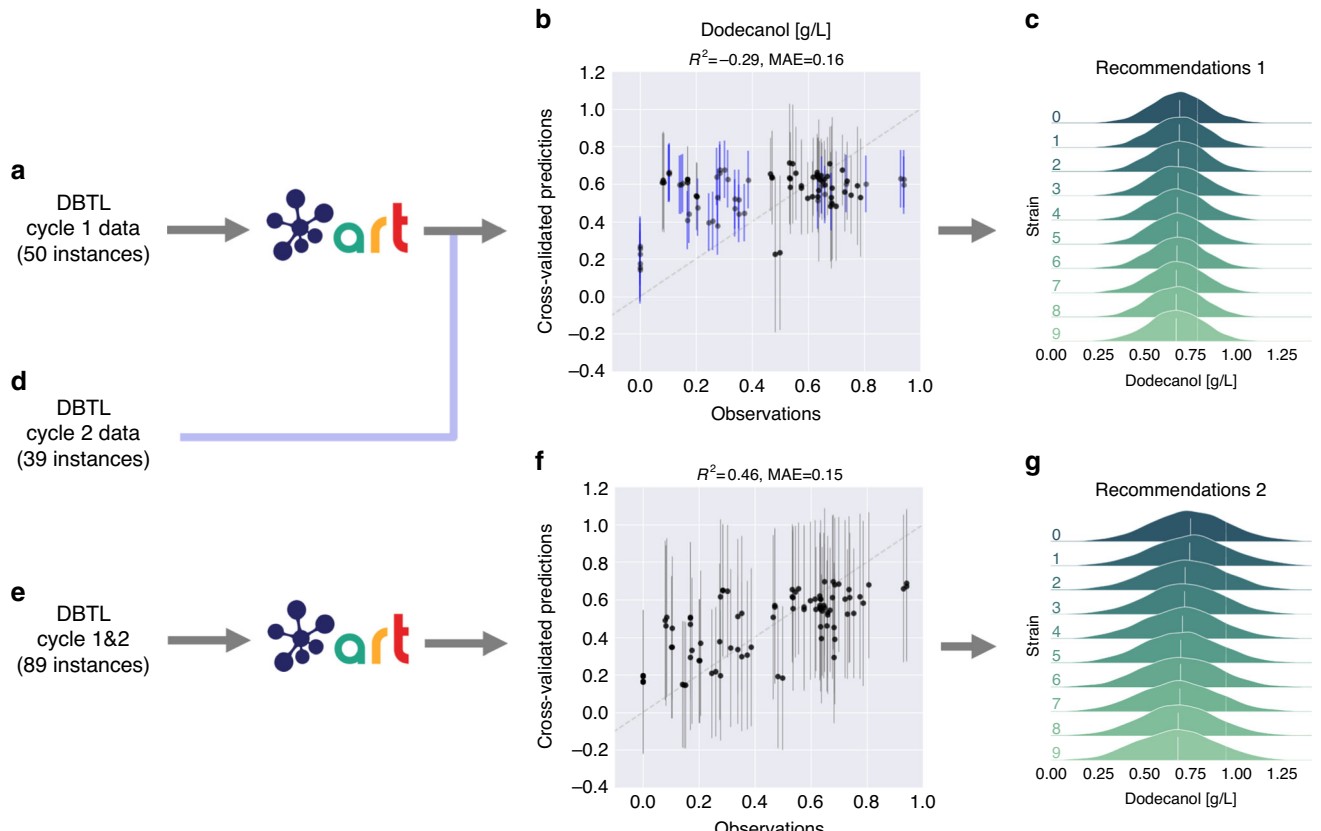

**Fig. 9 ART's predictive power is heavily compromised in the dodecanol case.** Although the 50 instances available for cycle 1 of pathway 1 (**a**) almost double the 27 available instances for the limonene case (Fig. 6), the predictive power of ART is heavily compromised ($R^2 = -0.29$ for cross-validation, **b**) by the scarcity of data and, we hypothesize, the strong tie of the pathway to host metabolism (fatty acid production). The poor predictions for the test data from cycle 2 (in blue) confirm the lack of predictive power. Adding data from both cycles (**d**, **e**) improves predictions notably (**f**). These data and model refer to the first pathway in Fig. 1B from ref. [41]. The cases for the other two pathways produce similar conclusions (Supplementary Figs. 8 and 9). Recommendations provided in panels **c** and **g**. $R^2$ and mean absolute error (MAE) values are only for cross-validated mean predictions (black data points). Bars indicate 95% credible interval of the predictive posterior distribution.

accuracy by leveraging a diverse set of models. This weighted ensemble model produces a simple, yet powerful, approach to quantify uncertainty (Fig. 6), a critical capability when dealing with small data sets and a crucial component of AI in biological research[60]. While ART is adapted to synthetic biology's special needs and characteristics, its implementation is general enough that it is easily applicable to other problems of similar characteristics. ART is perfectly integrated with the Experiment Data Depot[45] and the Inventory of Composable Elements[61], forming part of a growing family of tools that standardize and democratize synthetic biology.

We have showcased the use of ART in a case with synthetic data sets, three real metabolic engineering cases from the published literature, and a final case where ART is used to guide a bioengineering effort to improve productivity. The synthetic data case involves data generated for several production landscapes of increasing complexity and dimensionality. This case allowed us to test ART for different levels of difficulty of the production landscape to be learnt by the algorithms, as well as different numbers of DBTL cycles. We have seen that while easy landscapes provide production increases readily after the first cycle, more complicated ones require >5 cycles to start producing satisfactory results (Fig. 5). In all cases, results improved with the number of DBTL cycles, underlying the importance of designing experiments that continue for ~10 cycles rather than halting the project if results do not improve in the first few cycles.

The demonstration cases using previously published real data involve engineering *E. coli* and *S. cerevisiae* to produce the renewable biofuel limonene, synthesize metabolites that produce hoppy flavor in beer, and generate dodecanol from fatty acid biosynthesis. Although we were able to produce useful recommendations with as low as 27 (limonene, Fig. 6) or 19 (hopless beer, Fig. 8) instances, we also found situations in which larger amounts of data (50 instances) were insufficient for meaningful predictions (dodecanol, Fig. 9). It is impossible to determine a priori how much data will be necessary for accurate predictions, since this depends on the difficulty of the relationships to be learnt (e.g., the amount of coupling between the studied pathway and host metabolism). However, one thing is clear—two DBTL cycles (which was as much as was available for all these examples) are rarely sufficient for guaranteed convergence of the learning process. We do find, though, that accurate quantitative predictions are not required to effectively guide bioengineering—our ensemble approach can successfully leverage qualitative agreement between the models in the ensemble to compensate for the lack of accuracy (Fig. 7). Uncertainty quantification is critical to gauge the reliability of the predictions (Fig. 6), anticipate the smoothness of the recommendation process (Supplementary Figs. 6 and 7), and effectively guide the recommendations towards the least understood part of the phase space (exploration case, Fig. 3). We have also explored several ways in which the current approach (mapping proteomics data to production) can

fail when the underlying assumptions break down. Among the possible pitfalls is the possibility that recommended target protein profiles cannot be accurately reached, since the tools to produce specified protein levels are still imperfect; or because of biophysical, toxicity or regulatory reasons. These areas need further investment in order to accelerate bioengineering and make it more reliable, hence enabling the design to a desired specification. Also, it is highly recommendable to invest time in part characterization, pathway modularization, and experimental design to fully maximize the effectiveness of ART, and data-driven approaches in general (see the "Designing optimal experiments for machine learning" section in Supplementary Information for more details).

ART has also been used to guide metabolic engineering efforts to improve tryptophan productivity in yeast, as shown in the experimental counterpart of this publication[42]. In this project, genome-scale models were used to pinpoint which reactions needed optimization in order to improve tryptophan production. ART was then leveraged to choose which promoter combinations for the five chosen reactions would increase productivity. ART's recommendations resulted in a 106% increase in a productivity proxy with respect to the initial base strain. We would expect further increases if more DTBL cycles were to be applied beyond the initial two (see the "Using simulated data to test ART" section). This project showcases how ART can successfully guide bioengineering processes to increase productivity, a critical process metric[62] for which few systematic optimization methods exist. Furthermore, this project also demonstrates a case in which genetic parts (promoters) are recommended, instead of proteomics profiles as we did in the current paper. This approach has the advantage that it fully bridges the Learn and Design phases of the DBTL cycle, but it has the disadvantage that it may not fully explore the protein phase space (e.g., in case all promoters available are weak for a given protein).

Although ART is a useful tool in guiding bioengineering, it represents just an initial step in applying machine learning to synthetic biology. Future improvements under consideration include adding a pathway cost ($) function, classification problems, new optimization methods (e.g., to include the case of discrete input variables), the covariance of level-0 models into the ensemble model, input space errors into learners, and previous biological knowledge. These may not be the preferred list of improvements for every user, so ART's dual license allows for modification by third parties for research purposes, as long as the modifications are offered to the original repository. Hence, users are encouraged to enhance it in ways that satisfy their needs. A commercial use license is also available (see below for details).

ART provides effective decision-making in the context of synthetic biology and facilitates the combination of machine learning and automation that might disrupt synthetic biology[25]. Combining ML with recent developments in macroscale lab automation[47,63], microfluidics[38,64–66], and cloud labs[67] may enable self-driving laboratories[43,44], which augment automated experimentation platforms with artificial intelligence to facilitate autonomous experimentation. We believe that fully leveraging AI and automation can catalyze a similar step forward in synthetic biology as CRISPR-enabled genetic editing, high-throughput multi-omics phenotyping, and exponentially growing DNA synthesis capabilities have produced in the recent past.

## Methods

**Learning from data via a novel Bayesian ensemble approach.** Model selection is a significant challenge in machine learning, since there is a large variety of models available for learning the relationship between response and input, but none of them is optimal for all learning tasks[68]. Furthermore, each model features hyperparameters (i.e., parameters that are set before the training process) that crucially affect the quality of the predictions (e.g., number of trees for random forest or degree of polynomials in polynomial regression), and finding their optimal values is not trivial.

We have sidestepped the challenge of model selection by using an ensemble model approach. This approach takes the input of various different models and has them "vote" for a particular prediction. Each of the ensemble members is trained to perform the same task and their predictions are combined to achieve an improved performance. The examples of the random forest[69] or the super learner algorithm[70] have shown that simple models can be significantly improved by using a set of them (e.g., several types of decision trees in a random forest algorithm). Ensemble models typically either use a set of different models (heterogeneous case) or the same models with different parameters (homogeneous case). We have chosen a heterogeneous ensemble learning approach that uses reasonable hyperparameters for each of the model types, rather than specifically tuning hyperparameters for each of them.

ART uses a probabilistic ensemble approach where the weight of each ensemble model is considered a random variable, with a probability distribution inferred from the available data. Unlike other approaches[71–74], this method does not require the individual models to be probabilistic in nature, hence allowing us to fully exploit the popular scikit-learn library to increase accuracy by leveraging a diverse set of models (see "Related work and novelty of our ensemble approach" in Supplementary Information). Our weighted ensemble model approach produces a simple, yet powerful, way to quantify both epistemic and aleatoric uncertainty—a critical capability when dealing with small data sets and a crucial component of AI in biological research[60]. Here we describe our approach for the single response variable problems, whereas the multiple variables case can be found in the "Multiple response variables" section in Supplementary Information. Using a common notation in ensemble modeling we define the following levels of data and learners (Fig. 2):

- *Level-0 data* ($\mathcal{D}$) represent the historical data consisting of $N$ known instances of inputs and responses:

$$\mathcal{D} = \{(\mathbf{x}_n, y_n), n = 1, \ldots, N\}, \quad \mathbf{x} \in \mathcal{X} \subseteq \mathbb{R}^D, \quad y \in \mathbb{R}, \qquad (1)$$

  where $\mathbf{x}$ is the input comprised of $D$ features ($\mathcal{X}$ is the input phase space, Fig. 1) and $y$ is the associated response variable. For the sake of cross-validation, the *level-0 data* are further divided into validation ($\mathcal{D}^{(k)}$) and training sets ($\mathcal{D}^{(-k)}$). $\mathcal{D}^{(k)} \subset \mathcal{D}$ is the $k$th fold of a $K$-fold cross-validation obtained by randomly splitting the set $\mathcal{D}$ into $K$ almost equal parts, and $\mathcal{D}^{(-k)} = \mathcal{D} \setminus \mathcal{D}^{(k)}$ is the set $\mathcal{D}$ without the $k$th fold $\mathcal{D}^{(k)}$. Note that these sets do not overlap and cover the full available data; i.e., $\mathcal{D}^{(k_i)} \cap \mathcal{D}^{(k_j)} = \emptyset, i \neq j$ and $\cup_i \mathcal{D}^{(k_i)} = \mathcal{D}$.

- *Level-0 learners* ($f_m$) consist of $M$ base learning algorithms $f_m$, $m = 1, \ldots, M$ used to learn from level-0 training data $\mathcal{D}^{(-k)}$. For ART, we have chosen the following eight algorithms from the scikit-learn library: Random Forest, Neural Network, Support Vector Regressor, Kernel Ridge Regressor, K-NN Regressor, Gaussian Process Regressor, Gradient Boosting Regressor, as well as TPOT (tree-based pipeline optimization tool[75]). TPOT uses genetic algorithms to find the combination of the 11 different regressors and 18 different preprocessing algorithms from scikit-learn that, properly tuned, provides the best achieved cross-validated performance on the training set (https://github.com/EpistasisLab/tpot/blob/master/tpot/config/regressor.py)[75].

- *Level-1 data* ($\mathfrak{D}_{CV}$) are data derived from $\mathcal{D}$ by leveraging cross-validated predictions of the level-0 learners. More specifically, level-1 data are given by the set $\mathfrak{D}_{CV} = \{(\mathbf{z}_n, y_n), n = 1, \ldots, N\}$, where $\mathbf{z}_n = (z_{1n}\ldots, z_{Mn})$ are predictions for level-0 data ($\mathbf{x}_n \in \mathcal{D}^{(k)}$) of level-0 learners ($f_m^{(-k)}$) trained on observations which are not in fold $k$ ($\mathcal{D}^{(-k)}$), i.e., $z_{mn} = f_m^{(-k)}(\mathbf{x}_n), m = 1, \ldots, M$.

- The *level-1 learner* ($F$), or metalearner, is a linear weighted combination of level-0 learners, with weights $w_m$, $m = 1, \ldots, M$ being random variables that are non-negative and normalized to one. Each $w_m$ can be interpreted as the relative importance of model $m$ in the ensemble. More specifically, given an input $\mathbf{x}$ the response variable $y$ is modeled as:

$$F : \quad y = \mathbf{w}^T \mathbf{f}(\mathbf{x}) + \varepsilon, \quad \varepsilon \sim \mathcal{N}(0, \sigma^2), \qquad (2)$$

  where $\mathbf{w} = [w_1 \ldots w_M]^T$ is the vector of weights such that $\Sigma w_m = 1$, $w_m \geq 0$, $\mathbf{f}(\mathbf{x}) = [f_1(\mathbf{x}) \ldots f_M(\mathbf{x})]^T$ is the vector of level-0 learners, and $\varepsilon$ is a normally distributed error variable with a zero mean and standard deviation $\sigma$. The constraint $\Sigma w_m = 1$ (i.e., that the ensemble is a convex combination of the base learners) is empirically motivated but also supported by theoretical considerations[76]. We denote the unknown ensemble model parameters as $\theta \equiv (\mathbf{w}, \sigma)$, constituted of the vector of weights and the Gaussian error standard deviation. The parameters $\theta$ are obtained by training $F$ on the level-1 data $\mathfrak{D}_{CV}$ only. However, the final model $F$ to be used for generating predictions for new inputs uses these $\theta$, inferred from level-1 data $\mathfrak{D}_{CV}$, and the base learners $f_m$, $m = 1, \ldots, M$ trained on the full original data set $\mathcal{D}$, rather than only on the level-0 data partitions $\mathcal{D}^{(-k)}$. This follows the usual procedure in developing ensemble learners[77,78] in the context of stacking[76].

Rather than providing a single point estimate of ensemble model parameters $\theta$ that best fit the training data, a Bayesian model provides a joint probability distribution

$p(\boldsymbol{\theta}|\mathcal{D})$, which quantifies the probability that a given set of parameters explains the training data. This Bayesian approach makes it possible to not only make predictions for new inputs but also examine the uncertainty in the model. Model parameters $\boldsymbol{\theta}$ are characterized by full posterior distribution $p(\boldsymbol{\theta}|\mathcal{D})$ that is inferred from level-1 data. Since this distribution is analytically intractable, we sample from it using the Markov Chain Monte Carlo (MCMC) technique[79], which samples the parameter space with a frequency proportional to the desired posterior $p(\boldsymbol{\theta}|\mathcal{D})$ (see the "Markov Chain Monte Carlo sampling" section in Supplementary Information).

The important point is that, as a result, instead of obtaining a single value as the prediction for the response variable, the ensemble model produces a full probabilistic distribution that takes into account the uncertainty in model parameters. More precisely, for a new input $\mathbf{x}^*$ (not present in $\mathcal{D}$), the ensemble model $F$ provides the probability that the response is $y$, when trained with data $\mathcal{D}$ (i.e., the full predictive posterior distribution):

$$p(y|\mathbf{x}^*, \mathcal{D}) = \int p(y|\mathbf{x}^*, \boldsymbol{\theta})p(\boldsymbol{\theta}|\mathcal{D})\mathrm{d}\boldsymbol{\theta} = \int \mathcal{N}(y; \boldsymbol{w}^T\mathbf{f}, \sigma^2)p(\boldsymbol{\theta}|\mathcal{D})\mathrm{d}\boldsymbol{\theta}. \quad (3)$$

where $p(y|\mathbf{x}^*, \boldsymbol{\theta})$ is the predictive distribution of $y$ given input $\mathbf{x}^*$ and model parameters $\boldsymbol{\theta}$ (Eq. (2)), $p(\boldsymbol{\theta}|\mathcal{D})$ is the posterior distribution of model parameters given data $\mathcal{D}$, and $\mathbf{f} \equiv \mathbf{f}(\mathbf{x}^*)$ for the sake of clarity.

ART is very different from, and produces more accurate predictions than, Gaussian processes, a commonly used machine learning approach for outcome prediction and new input recommendations[43,80]. ART and Gaussian process regression[81] (GPR) share their probabilistic nature, i.e., the predictions for both methods are probabilistic. However, for GPR the prediction distribution is always assumed to be a Gaussian, whereas ART does not assume any particular form of the distribution and provides the full probability distribution (more details can be found in the "Expected value and variance for ensemble model" section of Supplementary Information). Moreover, ART is an ensemble model that includes GPR as one of the base learners for the ensemble. Hence, ART will, by construction, be at least as accurate as GPR (ART typically outperforms all its base learners). As a downside, ART requires more computations than GPR, but this is not a problem with the small data sets typically encountered in synthetic biology.

**Optimization-suggesting next steps.** The optimization phase leverages the predictive model described in the previous section to find inputs that are predicted to bring us closer to our objective (i.e., maximize or minimize response, or achieve a desired response level). In mathematical terms, we are looking for a set of $N_r$ suggested inputs $\mathbf{x}_r \in \mathcal{X}; r = 1, \ldots, N_r$, that optimize the response with respect to the desired objective. Specifically, we want a process that:

i. optimizes the predicted levels of the response variable;
ii. can explore the regions of input phase space ($\mathcal{X}$ in Eq. (1)) associated with high uncertainty in predicting response, if desired;
iii. provides a set of different recommendations, rather than only one.

We are interested in exploring regions of input phase space associated with high uncertainty, so as to obtain more data from that region and improve the model's predictive accuracy. Several recommendations are desirable because several attempts increase the chances of success, and most experiments are done in parallel for several conditions/strains.

In order to meet these three requirements, we define the optimization problem formally as

$$\underset{\mathbf{x}}{\arg\max} \, G(\mathbf{x}) \\ \text{s.t.} \, \mathbf{x} \in \mathcal{B} \quad (4)$$

where the surrogate function $G(\mathbf{x})$ is defined as:

$$G(\mathbf{x}) = \begin{cases} (1-\alpha)\mathbb{E}(y) + \alpha\mathrm{Var}(y)^{1/2} & \text{(maximization case)} \\ -(1-\alpha)\mathbb{E}(y) + \alpha\mathrm{Var}(y)^{1/2} & \text{(minimization case)} \\ -(1-\alpha)||\mathbb{E}(y) - y^*||_2^2 + \alpha\mathrm{Var}(y)^{1/2} & \text{(specification case)} \end{cases} \quad (5)$$

depending on which mode ART is operating in (see the "Key capabilities" section). Here, $y^*$ is the target value for the response variable, $y = y(\mathbf{x})$, $\mathbb{E}(y)$ and $\mathrm{Var}(y)$ denote the expected value and variance, respectively (see "Expected value and variance for ensemble model" in Supplementary Information), $||\mathbf{x}||_2^2 = \sum_i x_i^2$ denotes Euclidean distance, and the parameter $\alpha \in [0, 1]$ represents the exploitation-exploration trade-off (see below). The constraint $\mathbf{x} \in \mathcal{B}$ characterizes the lower and upper bounds for each input feature (e.g., protein levels cannot increase beyond a given, physical, limit). These bounds can be provided by the user (see details in the "Implementation" section in Supplementary Information); otherwise, default values are computed from the input data as described in the "Input space set $\mathcal{B}$" section in Supplementary Information.

Requirements (i) and (ii) are both addressed by borrowing an idea from Bayesian optimization[82]: optimization of a parametrized surrogate function which accounts for both exploitation and exploration. Namely, our objective function $G(\mathbf{x})$ takes the form of the upper confidence bound[83] given in terms of a weighted sum of the expected value and the variance of the response (parametrized by $\alpha$, Eq. (5)). This scheme accounts for both exploitation and exploration: for the

maximization case, for example, for $\alpha = 1$, we get $G(\mathbf{x}) = \mathrm{Var}(y)^{1/2}$, so the algorithm suggests next steps that maximize the response variance, thus exploring parts of the phase space where our model shows high predictive uncertainty. For $\alpha = 0$, we get $G(\mathbf{x}) = E(y)$, and the algorithm suggests next steps that maximize the expected response, thus exploiting our model to obtain the best response. Intermediate values of $\alpha$ produce a mix of both behaviors. In our example below using ART on simulated data we use a tempering strategy for $\alpha$ (see the "Using simulated data to test ART" section), whereas in the three experimental cases we set $\alpha = 0$. In general, we recommend setting $\alpha$ to values slightly smaller than one for early-stage DBTL cycles, thus allowing for more systematic exploration of the space so as to build a more accurate predictive model in the subsequent DBTL cycles. If the objective is purely to optimize the response, we recommend setting $\alpha = 0$.

In order to address (iii), as well as to avoid entrapment in local optima and search the phase space more effectively, we choose to solve the optimization problem through sampling. More specifically, we draw samples from a target distribution defined as

$$\pi(\mathbf{x}) \propto \exp(G(\mathbf{x}))p(\mathbf{x}), \quad (6)$$

where $p(\mathbf{x}) = \mathcal{U}(\mathcal{B})$ can be interpreted as the uniform "prior" on the set $\mathcal{B}$, and $\exp(G(\mathbf{x}))$ as the 'likelihood' term of the target distribution. Sampling from $\pi$ implies optimization of the function $G$ (but not reversely), since the modes of the distribution $\pi$ correspond to the optima of $G$. As we did before, we resort to MCMC for sampling. The target distribution is not necessarily differentiable and may well be complex. For example, if it displays more than one mode, as is often the case in practice, there is a risk that a Markov chain gets trapped in one of them. In order to make the chain explore all areas of high probability one can "flatten/melt down" the roughness of the distribution by tempering. For this purpose, we use the parallel-tempering algorithm[84] for optimization of the objective function through sampling, in which multiple chains at different temperatures are used for exploration of the target distribution (Fig. 3).

Choosing recommendations for the next cycle is done by selecting from the sampled inputs. After drawing a certain number of samples from $\pi(\mathbf{x})$ we need to choose recommendations for the next cycle, making sure that they are sufficiently different from each other as well as from the input experimental data. To do so, first we find a sample with optimal $G(\mathbf{x})$ (note that $G(\mathbf{x})$ values are already calculated and stored). We only accept this sample as a recommendation if there is at least one feature whose value is different by at least a factor $\gamma$ (e.g., 20% difference, $\gamma = 0.2$) from the values of that feature in all data points $\mathbf{x} \in \mathcal{D}$. Otherwise, we find the next optimal sample and check the same condition. This procedure is repeated until the desired number of recommendations are collected, and the condition involving $\gamma$ is satisfied for all previously collected recommendations and all data points. In case all draws are exhausted without collecting the sufficient number of recommendations, we decrease the factor $\gamma$ and repeat the procedure from the beginning. Pseudo code for this algorithm can be found in Algorithm 1 in Supplementary Information. The probability of success for these recommendations is computed as indicated in the "Success probability calculation" section in Supplementary Information.

**Implementation.** ART is implemented Python 3.6 and should be used under this version (see below for software availability). Supplementary Fig. 1 represents the main code structure and its dependencies to external packages. In the "Implementation" section of Supplementary Information, we provide an explanation for the main modules and their functions.

**Reporting summary.** Further information on research design is available in the Nature Research Reporting Summary linked to this article.

## Data availability

A reporting summary for this Article is available as a Supplementary Information file. The experimental data can also be found in the Experiment Data Depot[45], in the following studies: Study Link Biofuel: https://public-edd.jbei.org/s/pcap/; Hopless beer: https://public-edd.agilebiofoundry.org/s/hopless-beer/, https://public-edd.agilebiofoundry.org/s/hopless-beer-cycle-2/; Dodecanol: https://public-edd.jbei.org/s/ajinomoto/. Freely available accounts on public-edd.jbei.org and public-edd.agilebiofoundry.org are required to view and download these studies. Source data are provided with this paper.

## Code availability

ART is developed under Python 3.6 and relies on packages: seaborn 0.7.1, scikit-learn 0.20.2, pymc3 3.5, pandas 0.23.4, numpy 1.14.3, matplotlib 3.0.2, scipy 1.1.0, PTMCMC Sampler 2015.2. ART's dual license allows for free non-commercial use for academic institutions. Modifications should be fed back to the original repository for the benefit of all users. A separate commercial use license is available from Berkeley Lab (ipo@lbl.gov). See https://github.com/JBEI/ART for software and licensing details.

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

## Acknowledgements

This work was part of the Agile BioFoundry (http://agilebiofoundry.org) and the DOE Joint BioEnergy Institute (http://www.jbei.org), supported by the U. S. Department of Energy, Energy Efficiency and Renewable Energy, Bioenergy Technologies Office, and the Office of Science, through contract DE-AC02-05CH11231 between Lawrence Berkeley National Laboratory and the U.S. Department of Energy. The United States Government retains and the publisher, by accepting the article for publication, acknowledges that the United States Government retains a nonexclusive, paid-up, irrevocable, worldwide license to publish or reproduce the published form of this manuscript, or allow others to do so, for United States Government purposes. The Department of Energy will provide public access to these results of federally sponsored research in accordance with the DOE Public Access Plan (http://energy.gov/downloads/doe-public-access-plan). This research is also supported by the Basque Government through the BERC 2014-2017 program and by Spanish Ministry of Economy and Competitiveness MINECO: BCAM Severo Ochoa excellence accreditation SEV-2013-0323. We acknowledge and thank Peter St. John, Joonhoon Kim, Amin Zargar, Henrique De Paoli, Sean Peisert, and Nathan Hillson for reviewing the manuscript and for their helpful suggestions.

## Author contributions

Z.C., T.R., and H.G.M. conceived the original idea. T.R. and Z.C. developed a methodology, designed the software, wrote the code, and performed computer experiments. T.R. designed simulated benchmarks and performed numerical experiments. T.R. analyzed all results. K.W. wrote tests and documented the code. H.G.M. and T.R. wrote the paper.

## Competing interests

The authors declare no competing interests.
