## [Peer Review File · Nature Communications]

Reviewers' Comments:

Reviewer #1:

Remarks to the Author:

Summary

The manuscript from Radivojevic et al presents ART, a machine learning-based recommendation tool for metabolic engineering. The approach combines several ideas from machine learning such as ensemble modelling, Bayesian predictions with associated probability distribution and exploitation/exploration, which is interesting and has some novelty aspects. In particular, it provides a reasonable solution when dealing with the often-found problem of low availability of training data. However, all the results are either simulated or obtained from previous experimental data (DBTL cycles) which were not actually guided by ART recommendations and therefore presenting more solid results in the manuscript should be desirable.

Comments

P4I58 This paragraph about ML is probably unnecessary or could be just summarized to the few biological applications that are mentioned.

P4I68 Combining scikit-learn and Bayesian approach does not seem very innovative or even relevant for the Introduction.

P4I72 If data sets contain less than 100 instances, it is highly speculative assume that the approach will be able to integrate successfully deep learning capabilities. There is no doubt that the deep learning and scikit-learn can be easily integrated, but it is less clear what is the advantage that deep learning or ART can bring in.

P5I92 "successfully guide the bioengineering process even in the absences of quantitatively accurate predictions". This statement will need to be substantiated.

P6I99 First mention to "automation". Please define also what the authors mean by "inverse design".

P6I102 These last two sentences might sound too patronizing for the readers.

P6I116 It is not clear what the "EDD-style .csv files" are. A clear link to some online information or supplementary data has to be provided.

P6I120 "the full probability distribution of the predictions". Not clear what "full" means in this sentence.

Fig. 1 is not very informative. Left side is quite trivial, it is not clear why the Predictive model is not connected to the recommendations. Finally, the recommendations are similar to the left side but with some curve plot which is not explained.

P7I131 Mention to "beer taste profile" here is not well-justified.

P9I166 What are the similarities/differences of the probabilistic approach and Gaussian processes?

P10I186 Numbers are not clear: 8 algorithms, 11 regressors, and 18 preprocessing algorithms. What are those regressors and algorithms?

P10I194 Does TPOT find the optimal output combination of all learners? In that case, it would not be clear why again the predictions of each individual learner are used as inputs to the next level.

P11I200 I am not convince that a weight w_m can be interpreted as the "relative confidence in model m ". First, are the z predictions normalized? Otherwise the weights might be interpreted as some rescaling. Second, even if predictions are scaled, the weights would be interpreted as regularization factors, i.e. smoothing the prediction curves of the learners in order to minimize the variance.

P12I231 Even if the "mathematical methodology" seems sophisticated, at the end what counts is that predictions are given as a probability distribution.

P12I239 The goal of exploring "the regions of input phase space associated with high uncertainty in predicting reponse" should be better explained. What is the phase space in this context? Why should the user desire exploring high uncertainty regions? In order to refine the model through additional experimental data? It seems to be explained in next paragraphs (exploration), it is recommended to add the terms exploitation/exploration already to the requirements.

Fig. 3 Left panel shows the relationship between the response y and the input x . x is defined as an example as "proteomics data". For the example of proteomics data, what is the interpretation of

the "recommendations" (the blue dots). Generally, a proteomics value would not be "recommended" but rather measured. Do the authors mean that in the example the recommendations are about varying the expression levels of the proteins? This important point needs better discussion and context. The proteomics data are restricted to engineered proteins in the host?

P16I304 The simulation section is not very informative. It seems as if the authors are generating results that follow what already could be expected based on the mathematical approach. Basically, this section seems to be for debugging purposes. If x are proteomics data, how the E,M,D functions should be interpreted? Is there any reason that justifies the F_d formula? It would be more reasonable simulating some actual biological process, for instance biosynthetic pathways through their kinetic equations and their stationary proteomics data to the learning processes through several DBTL cycles.

P19I34 The introductory paragraphs about limonene production are written like a review paper. They should be summarized.

P20I378 The rationale of this section is not clear. We can assume that the second DBTL cycle was guided by the predictions obtained by the PCAP algorithm. Therefore, the fact that the proposed ART algorithm proposes the same recommendations does not show any clear improvement.

P20I387 A cross validation of $R^2 = 0.44$ is not very convincing.

P23I419 For the DBTL cycles of the hoppy beer example, it is not clear whether the authors mean that ART could not be trained (bad outcome on the test set) with the data of the first cycle, or conversely it was successfully trained and provided good recommendations. This issue is repeated at other parts of this section. For instance, p25I452 "despite the limited predictive power afforded by the cycle 1 data, ART recommendations guide metabolic engineering effectively". This sentence is too ambiguous.

P26I459 The conclusions of the third example (dodecanol production) are too speculative. If the algorithm was not able to learn because the problem is hard and there were not enough data available, I think that the authors should refrain from "blaming" biological mechanisms like cell membrane production, which might be linked or not to dodecanol production.

P27I495 Another issue, from the point of view of this reviewer, is that machine learning is used in order to learn the relationship between protein levels and production. However, any recommendation on levels obtained through the algorithm needs to be matched with the appropriate engineering (RBS, etc.) and therefore there is no direct method. In fact the authors mentioned cases where it was not possible to obtain the prescribed fold-increase in the protein expression.

P32 The limonene example (html) generated by a notebook cannot be visualized directly on GitHub. It would be better to store it as ipynb notebook.

P32 I was unable to access the hopless beer data link.

P32 Password-protected GitHub page does not seem to work.

Reviewer #2:

Remarks to the Author:

In their manuscript Radivojević et al. present ART: A machine learning Automated Recommendation Tool for Synthetic Biology. ART combines machine learning and probabilistic modelling techniques to guide synthetic biology projects towards their specific engineering goal. ART learns from a first set of experimental data and recommends new strains to be engineered over several DBTL cycles.

The manuscript provides a solution (or the beginning of a solution) for a very important current challenge in Synthetic Biology: Namely how we can improve the learning step within the typical DBTL cycles that are required in metabolic engineering and Synthetic Biology, such that optimally performing strains can be reached faster and in a more systematic fashion.

In the short-term this would indeed readily allow to reduce cost for the successful implementation of new Synthetic Biology projects and eventually – in the long-term – enable fully automated metabolic engineering/Synthetic Biology labs.

The authors test ART with three simulated engineering efforts of different difficulty and three real ME examples from literature. While ART readily gives satisfactory suggestions after a few DBTL cycles for "easy" ME problems, it requires significantly more DBTL cycles for more difficult ones. Still, even in difficult cases ART could guide decision making and incremental strain improvements, where – without machine learning help – a project would likely be abandoned after a few cycles. Most importantly ART shows that difficult ME projects can be realised step-by step, but only if the metabolic engineer is prepared to undergo >10 reengineering cycles. This is essential knowledge for decision making in difficult but potentially impactful projects and challenges current ME practice (only going through very few DBTL cycles).

As such I consider the developed ART tool and the presented results a major improvement over state-of-the-art metabolic engineering practice.

Still, from an experimentalists point of view the authors leave several points open or under-discussed which reduces the potential usefulness of this manuscript for a user that wants to incorporate ART into their next Synthetic Biology project. But these points can be addressed.

Major points

1. Experimental design of the starting (training) data: The manuscript does not examine or elaborate on the importance of properly designing the first set of input data. Given machine learning can provide predictive power by learning the underlying patterns in experimental data, I would assume that it is highly important that these data are as informative as possible from beginning on. E.g. starting data should sample as effectively as possible the parameter space to give the best impression of the underlying landscape.

As ART will eventually be used to guide new Synthetic Biology efforts, the experimentalist has some freedom in designing the starting data and needs advice on how to best do that (especially given that usually only low number of instances can be generated).

For example, in the original papers that were used in the current manuscript, the authors apply some level of experimental design (promoter strength, induction levels), but it is unclear if those considerations are actually optimal or useful for ART.

Denby et. al: "An initial set of 18 strains containing promoters predicted to span a wide range of expression strengths were constructed"

Alonso-Gutierrez et al.: "the enzymes in different gene clusters was generated through variation of promoter strength, different plasmid copy numbers, and under different induction timings and levels"

I suggest:

- The importance of effective experimental design and landscape sampling should be examined or illustrated by using different experimental designs in the simulated data case. e.g. testing the relation between "good" starting data and DBTL cycles needed.
- Further the experimental design should be discussed in the manuscript more thoroughly in form of guidelines or "things to consider" for experimentalists.

2. The format of the final recommendation: The manuscript does not illustrate how the eventual output data (strain recommendations) look like; but the format of this recommendation has practical implications for the a priori pathway design (e.g. in terms of modularity). I assume the output is a list of strains linked to a list of different relative concentrations of the pathway's components (e.g. enzymes). Are the new recommended relative concentrations set into the context of one or all (or none) of the previously existing strains?

I think a clear illustration (at least a supplementary figure) of the output data would help a potential biology user understand what type of suggestions ART gives and what type of fine-tuning is expected from a pathway. That would facilitate effective pathway design.

For example, the pathway needs to be sufficiently modular and the parts within it sufficiently characterised. As such, the usage and extension of e.g. MoClo toolboxes would be advisable.

The manuscript mentions issues with ineffective a priori design, missing modularity and unpredictable part behaviour (lines 497, 500 and 512) but these points should be focused and discussed in a separate section, ideally together with the experimental design (point 1). In that context it would be interesting to discuss how the use of machine learning in SynBio uncovers new (or let's say old but under-addressed) bottlenecks, like the need for well characterised parts/predictable part behaviour, insulated pathways, standardisations etc.

Minor points:

1. Definition of the term instances: while it becomes clearer towards the end of the manuscript (instances = refereeing to differently engineered strains and/or induction conditions thereof, that lead to different production readouts because of different expression level profiles; eventually used as a training set), the term instances should be defined when it is used for the first time e.g. in line 71 in the introduction or at least in line 310. In line 310 the sentence "with a training set given by 16 strains" lends itself to explain that those are referred to as instances.

In the conclusion section (line 517) the term instances seems to be replaced or reworded into "a set of vectors of measurements".

In summary, the terminology for the input data needs to be clarified and described more consistently in order for the biology community to get a clear picture what data are fed into ART and accordingly which data need to be experimentally produced to start the ART-assisted DBTL.

2. More careful usage of the term -omics data: The manuscript uses the term -omics data when describing the input data that are required for ART. I think the term -omics data is a bit misleading (in disfavour of the manuscript). While not being wrong, it gives the impression that full -omics profiles of engineered strains are required for ART, which would be highly costly. I would clarify that and use "gene expression data" or "targeted proteomic data" for the pathway components of interest.

3. All figures relating to the use of ART for real ME problems (5, 7 and 8) use the same graph type to illustrate the data: Observations are plotted against cross-validated predictions. One of the figure legends should be used to effectively explain this graph type to a biology user, how to interpret the data and why it is the best way to illustrate the data.

4. In figures 5,7 and 8 labelling of panels with a,b,c etc. instead of "top right" etc. would enhance readability of the figures.

5. Figure 6, the units for limonene production are missing.

Reviewer 1:

Summary

The manuscript from Radivojevic et al presents ART, a machine learning-based recommendation tool for metabolic engineering. The approach combines several ideas from machine learning such as ensemble modelling, Bayesian predictions with associated probability distribution and exploitation/exploration, which is interesting and has some novelty aspects. In particular, it provides a reasonable solution when dealing with the often-found problem of low availability of training data.

We thank the reviewer for the positive comments: in particular, the novelty aspects of the technique, and the fact that it provides a solution to the common problem of low availability of training data.

However, all the results are either simulated or obtained from previous experimental data (DBTL cycles) which were not actually guided by ART recommendations and therefore presenting more solid results in the manuscript should be desirable.

Actually, ART has been successfully used to guide metabolic engineering to improve tryptophan production. The results for these experiments are being considered for publication at this moment in this same journal (NCOMMS-19-42070) and can be found in the following preprint in biorxiv (as well as part of this submission):

<https://www.biorxiv.org/content/10.1101/858464v1>.

Our paper constitutes the theoretical counterpart to this experimental study, which is published separately, and in parallel, so as not to unnecessarily compress the findings and methods.

However, using simulated data is crucial because it allows us to explore scenarios that would be prohibitively expensive to test experimentally (e.g. ten DTBL cycles, different levels of difficulty), and provides useful information in terms of planning future experiments.

Moreover, leveraging experimental data from other experiments allows us to test ART with other pathways, molecules and objectives (e.g. specification vs maximization) than increasing tryptophan production in yeast, so as to show that ART is indeed a tool of general utility.

We cited the above paper and discussed these considerations in the introduction:

“The efficacy of ART in guiding synthetic biology is showcased through five different examples: one test case using simulated data, three cases where we leveraged previously collected experimental data from real metabolic engineering projects, and a final case where ART is used to guide a bioengineering effort to improve productivity. In the synthetic case and the three experimental cases where previous data is leveraged, we mapped one type of --omics data (proteomics in particular) to production. In the case of using ART to guide experiments, we mapped promoter combinations to production. In all cases the underlying assumption is that the input (--omics data or promoter combinations) is predictive of the response (final

production), and that we have enough control over the system so as to produce any new recommended input. The test case permits us to explore how the algorithm performs when applied to systems that present different levels of difficulty when being "learnt", as well as the effectiveness of using several DTBL cycles. The real metabolic engineering cases involve data sets from published metabolic engineering projects: renewable biofuel production³⁹, yeast bioengineering to recreate the flavor of hops in beer⁴⁰, and fatty alcohols synthesis⁴¹. These projects illustrate what to expect under different typical metabolic engineering situations: high/low coupling of the heterologous pathway to host metabolism, complex/simple pathways, high/low number of conditions, high/low difficulty in learning pathway behavior. Finally, the fifth case uses ART in combination with genome-scale models to improve tryptophan productivity in yeast by 105% from the base strain, and is published in parallel⁴² as the experimental counterpart to this article. We find that ART's ensemble approach can successfully guide the bioengineering process even in the absence of quantitatively accurate predictions (see e.g. the "Improving the production of renewable biofuel" section). Furthermore, ART's ability to quantify uncertainty is crucial to gauge the reliability of predictions and effectively guide recommendations towards the least known part of the phase space. These experimental metabolic engineering cases also illustrate how applicable the underlying assumptions are, and what happens when they fail."

as well as the conclusion, to which we have added the following paragraph:

"ART has also been used to guide metabolic engineering efforts to improve tryptophan productivity in yeast, as shown in the experimental counterpart of this publication⁴². In this project, genome-scale models were used to pinpoint which reactions needed optimization in order to improve tryptophan production. ART was then leveraged to choose which promoter combinations for the five chosen reactions would increase productivity. ART's recommendations resulted in a 105% increase in productivity with respect to the initial base strain. We would expect further increases if more DTBL cycles were to be applied beyond the initial two (see the "Using simulated data to test ART" section). This project showcases how ART can successfully guide bioengineering processes to increase productivity, a critical process metric⁸¹ for which few systematic optimization methods exist. Furthermore, this project also demonstrates a case in which genetic parts (promoters) are recommended, instead of proteomics profiles as we did in the current paper. This approach has the advantage that it fully bridges the Learn and Design phases of the DBTL cycle, but it has the disadvantage that it may not fully explore the protein phase space (e.g. in case all promoters available are weak for a given protein)."

Comments

P4I58 This paragraph about ML is probably unnecessary or could be just summarized to the few biological applications that are mentioned.

While we intended to explain that ML can provide predictions without the need to understand the underlying mechanisms, the current paragraph does not do a good job at it. We apologize.

We have changed the paragraph to emphasize this point and have limited the associated

applications to the ones that support it. We also find it important to mention some of the recent successes of ML to the synthetic biology audience, who may not be fully aware of the state of the art in the field:

“Machine learning (ML) arises as an effective tool to predict biological system behavior and empower the Learn phase, enabled by emerging high-throughput phenotyping technologies²⁵. Machine learning can provide predictions without the need to understand the underlying mechanisms: e.g. it has been used to predict the use of addictive substances and political views from Facebook profiles²⁶, automate language translation²⁷, predict pathway dynamics²⁸, optimize pathways through translational control²⁹, diagnose skin cancer³⁰, detect tumors in breast tissues³¹, predict DNA and RNA protein-binding sequences³² and drug side effects³³. However, the practice of machine learning requires statistical and mathematical expertise that is scarce and highly competed for throughout industry and academia.³⁴”

P4I68 Combining scikit-learn and Bayesian approach does not seem very innovative or even relevant for the Introduction.

We agree with the reviewer that just considering the combination of scikit-learn and a Bayesian approach is not innovative. However, the practical approach we used for this combination is novel, simple, and effective (see “Related work and novelty of our ensemble approach” section in supp. material). Furthermore, this combination is at the crux of this paper, since it enables quantification of uncertainty, and provides a systematic approach to leverage increasing amounts of data. Hence, we believe this combination is very relevant for the introduction and have chosen to keep it in its current form.

P4I72 If data sets contain less than 100 instances, it is highly speculative to assume that the approach will be able to integrate successfully deep learning capabilities. There is no doubt that the deep learning and scikit-learn can be easily integrated, but it is less clear what is the advantage that deep learning or ART can bring in.

Our apologies for the misunderstanding. We meant to say that with the usual synthetic biology data sets involving less than 100 instances, deep learning is of limited utility. However, once high-throughput data generation and automated data collection capabilities are widespread, we expect that data sets of thousands, tens of thousands and even more instances will be customarily available, and deep learning can be profitably leveraged. Even in that case, the novel ART Bayesian approach will be useful in the context of deep learning. We have changed the text accordingly:

“The data sets collected in the synthetic biology field (<100 instances) are typically not large enough to allow for the use of deep learning, so this technique is not currently used in ART. However, once high-throughput data generation^{14,37} and automated data collection³⁸

capabilities are widely used, we expect data sets of thousands, tens of thousands, and even more instances to be customarily available, enabling deep learning capabilities that can also leverage ART's Bayesian approach. In general, ART provides machine learning capabilities in an easy-to-use and intuitive manner, and is able to guide synthetic biology efforts in an effective way.”

P5I92 “successfully guide the bioengineering process even in the absences of quantitatively accurate predictions”. This statement will need to be substantiated.

Indeed. We believe that this statement has been substantiated in Fig. 6, as well as in the parallel experimental counterpart paper (NCOMMS-19-42070), and, to a lesser extent, in Fig. S7. We have included this consideration in the caption for Fig. 6:

Although the models differ significantly in the actual quantitative predictions of production, the same qualitative trends can be seen in all models (i.e. explore upper right quadrant for higher production), justifying the ensemble approach used by ART. The ART recommendations (green) are very close to the protein profiles from the PCAP paper (red), that were experimentally tested to improve production by 40%. Hence, we see that ART can successfully guide the bioengineering process even in the absence of quantitatively accurate predictions.

and in the main text:

“Interestingly, we see that while the quantitative predictions of each of the individual models were not very accurate, they all signaled towards the right direction in order to improve production, showing the importance of the ensemble approach (Fig. 6). Hence, we see that ART can successfully guide the bioengineering process even in the absence of quantitatively accurate predictions.”

At any rate, despite the limited predictive power afforded by the cycle 1 data, ART recommendations guide metabolic engineering effectively. For both of the Pale Ale and Torpedo cases, ART recommends exploring parts of the proteomics phase space that surround the final protein profiles, which were deemed close enough to the desired targets in the original publication. Recommendations from cycle 1 and initial data (green and red in Fig. S7) surround the final protein profiles obtained in cycle 2 (orange in Fig. S7). Finding the final protein target becomes, then, an interpolation problem, which is much easier to solve than an extrapolation one.

“Finally, the fifth case uses ART in combination with genome-scale models to improve tryptophan productivity in yeast by 105% from the base strain, and is published in parallel⁴² as the experimental counterpart to this article.”

P6I99 First mention to “automation”. Please define also what the authors mean by “inverse design”.

“Inverse design” refers to the capability to design DNA to fit a specification. If we assume the DNA → phenotype mapping to be the forward problem, the capability of inverting the mapping (phenotype → DNA) to design DNA that meets a given phenotype is what is usually called inverse design. We have included this explanation in the text:

“inverse design (i.e. the capability to design DNA to meet a specified phenotype: a biofuel production rate, for example)”

P6I102 These last two sentences might sound too patronizing for the readers.

Thanks for the feedback. We have changed the wording to avoid any semblance of patronization:

“We have made a special effort to write this paper to be accessible to both the machine learning and synthetic biology readership, with the intention of providing a much needed bridge between these two very different collectives. Hence, we have emphasized explaining basic machine learning and synthetic biology concepts, since they might be of use to a part of the readership.”

P6I116 It is not clear what the “EDD-style .csv files” are. A clear link to some online information or supplementary data has to be provided.

We apologize for this oversight. The supplementary material does indeed contain information on what constitutes an EDD-style .csv file. We have now explicitly mentioned that in the text:

“Alternatively, ART can import EDD-style .csv files, which use the nomenclature and structure of EDD exported files (see the "Importing a study" section in the supplementary material).”

We also included an example of such a file in the section “Importing a study” in the supplementary material (Fig. S2).

P6I120 “the full probability distribution of the predictions”. Not clear what “full” means in this sentence.

By “the full probability distribution of the predictions”, we refer to the distribution that

represents the whole available knowledge—all of the possible outcomes for our response variable and their associated probability values. This is in contrast to maximum probability or expectation type of estimates, or estimates in which no probabilistic modeling was undertaken whatsoever. We have now clarified this in the following text in the main manuscript:

“Rather than predicting point estimates of the response variable, ART provides the full probability distribution of the predictions (i.e., the distribution for all possible outcomes for the response variable and their associated probability values).”

Fig. 1 is not very informative. Left side is quite trivial, it is not clear why the Predictive model is not connected to the recommendations. Finally, the recommendations are similar to the left side but with some curve plot which is not explained.

Figure 1 constitutes the graphical summary of the paper and the tool. While it may not be very informative to the expert eye of the reviewer, we believe that it is fundamental to succinctly explain what ART is to a reader with less expertise in machine learning. For these reasons and because it helps dispel misunderstandings that the reviewer poses below, we have chosen to keep it, although adapted to reflect the reviewer’s comments: we have connected the predictive model to the recommendations (since the former enables the latter), and we have further explained the curve plot (a probabilistic recommendation as in Fig. 5) and the difference between the inputs and recommendations.

P7I131 Mention to “beer taste profile” here is not well-justified.

The “beer taste profile” expression provides a very specific example of a bioengineering-relevant specification objective: “as well as specification objectives (e.g. to reach a specific level of a target molecule for a desired beer taste profile)”. We believe this very specific example dispels any question about whether the specification goal is relevant for bioengineering and ART. For clarity, we have now added the reference for this specific example (Fig. 7):

“specification objectives (e.g. to reach a specific level of a target molecule for a desired beer taste profile⁴⁰)”

P9I166 What are the similarities/differences of the probabilistic approach and Gaussian processes?

ART’s probabilistic approach is substantially different from a Gaussian process. We have included the following paragraph at the end of the “Learning from data: a predictive model through machine learning and a novel Bayesian ensemble approach” section to clarify this point:

“ART is very different from, and produces more accurate predictions than, Gaussian processes, a commonly used machine learning approach for outcome prediction and new input recommendations.^{43,60} ART and Gaussian process regression (GPR⁶¹) share their probabilistic nature: i.e., the predictions for both methods are probabilistic. However, for GPR the prediction distribution is always assumed to be a Gaussian, whereas ART does not assume any particular form of the distribution and provides the full probability distribution (more details can be found in the “Expected value and variance for ensemble model” section of the supplementary material). Moreover, ART is an ensemble model that includes GPR as one of the base learners for the ensemble. Hence, ART will, by construction, be at least as accurate as GPR (ART typically outperforms all its base learners). As a downside, ART requires more computations than GPR, but this is not usually a problem with the small data sets typically encountered in synthetic biology.”

P10I186 Numbers are not clear: 8 algorithms, 11 regressors, and 18 preprocessing algorithms. What are those regressors and algorithms?

We have now included a citation that provides the details for the regressors and algorithms used by TPOT:

“TPOT uses genetic algorithms to find the combination of the 11 different regressors and 18 different preprocessing algorithms from scikit-learn that, properly tuned, provides the best achieved cross-validated performance on the training set^{54,55}.”

P10I194 Does TPOT find the optimal output combination of all learners? In that case, it would not be clear why again the predictions of each individual learner are used as inputs to the next level.

Not exactly. TPOT finds the optimal combination of *one* of the learners with a set of preprocessing algorithms, rather than the optimal combination of all the learners. This optimization step, based on genetic algorithms, depends on its own hyperparameters (e.g. number of generations, population size) and might not necessarily result in the best predictive accuracy across different data sets. In contrast, ART uses the predictions of each learner to create an ensemble, which has been shown to be a superior approach. See, for example:

Breiman, L. Stacked regressions. Machine Learning 1996,24, 49–64

P11I200 I am not convinced that a weight w_m can be interpreted as the “relative confidence in model m ”. First, are the z predictions normalized? Otherwise the weights might be interpreted as some rescaling. Second, even if predictions are scaled, the weights would be interpreted as regularization factors, i.e. smoothing the prediction curves of the learners in order to minimize

the variance.

No, the z predictions are not normalized. We agree with the reviewer that the term “relative confidence” might not be the most appropriate. Given the constraint that weights are positive and sum to one, the ensemble prediction for response y is modeled as a weighted average of predictions from base learners. Therefore, the value of a weight associated to a learner m can be reasonably interpreted as the relative importance of the learner m in the ensemble. We have introduced modifications in the manuscript to reflect this interpretation:

“Each w_m can be interpreted as the relative importance of model m in the ensemble.”

Indeed, the fact that we are imposing a certain prior on weights (which ensures the constraints are satisfied) can be interpreted as a regularization, which in this case is Bayesian in nature. Nevertheless, we prefer to keep the more intuitive interpretation of relative importance for the weights.

P12I231 Even if the “mathematical methodology” seems sophisticated, at the end what counts is that predictions are given as a probability distribution.

We completely agree with the reviewer. We have now emphasized that the important point is that predictions are given as a probability distribution:

“The important point is that, as a result, instead of obtaining a single value as the prediction for the response variable, the ensemble model produces a full probabilistic distribution that takes into account the uncertainty in model parameters.”

P12I239 The goal of exploring “the regions of input phase space associated with high uncertainty in predicting response” should be better explained. What is the phase space in this context? Why should the user desire exploring high uncertainty regions? In order to refine the model through additional experimental data? It seems to be explained in next paragraphs (exploration), it is recommended to add the terms exploitation/exploration already to the requirements.

We have further explained in previous paragraphs the meaning of phase space, and in the next one why exploring high uncertainty regions is desirable in the caption of Figure 1:

“The input phase space, in this case, is composed by all the possible combinations of protein expression (or transcription levels, promoters, etc. for other cases).”

Equation 1:

“where x is the input comprised of D features (X is the input phase space, see Fig. 1) and y is the associated response variable.”

and as a new paragraph in the “Optimization: suggesting next steps” section:

“We are interested in exploring regions of input phase space associated with high uncertainty, so as to obtain more data from that region and improve the model's predictive accuracy. Several recommendations are desirable because several attempts will increase the chances of success and most experiments are done in parallel for several conditions/strains.”

Fig. 3 Left panel shows the relationship between the response y and the input x . x is defined as an example as “proteomics data”. For the example of proteomics data, what is the interpretation of the “recommendations” (the blue dots). Generally, a proteomics value would not be “recommended” but rather measured. Do the authors mean that in the example the recommendations are about varying the expression levels of the proteins? This important point needs better discussion and context. The proteomics data are restricted to engineered proteins in the host?

The recommendations are the suggested inputs for the next experiment (“Recommendations for next cycle” in Fig. 1) so as to obtain the desired goal (e. g. increase production). Inputs such as protein levels are first *recommended* as targets for varying protein expression levels of proteins, then the metabolic engineer finds ways to reach those recommended targets in the Design phase of DBTL cycle two, and then protein expression levels are *measured* in the Test phase of DTBL cycle two, to be compared with the recommended targets. An example of this comparison can be seen in the limonene example (“Improving the production of renewable biofuel” section) and, specifically, in Fig. 6. The proteomics data are not restricted to engineered proteins in the host: it could be any protein. Eliminating this misunderstanding further justifies keeping Fig. 1 as a fundamental figure to explain ART's functioning (see response above).

We have clarified these points by linking the recommendations in Figure 3 to the recommendations in Figure 1, and providing a specific example through Figure 6:

“Figure 3: **ART chooses recommendations for next steps by sampling the modes of a surrogate function.** The leftmost panel shows the true response y (e.g. biofuel production to be optimized) as a function of the input x (e.g. proteomics data), as well as the expected response $E(y)$ after several DBTL cycles, and its 95% confidence interval (blue). Depending on whether we prefer to explore the phase space where the model is least accurate or exploit the predictive model to obtain the highest possible predicted responses, we will seek to optimize a surrogate function $G(x)$ (Eq. 5), where the exploitation-exploration parameter is $\alpha=0$ (pure exploitation), $\alpha=1$ (pure exploration) or anything in between. Parallel-Tempering-based MCMC sampling (center and right side) produces sets of vectors x (colored dots) for different

“temperatures”: higher temperatures (red) explore the full phasespace, while lower temperature chains (blue) concentrate in the nodes (optima) of $G(x)$. Exchange between different “temperatures” provides more efficient sampling without getting trapped in local optima. Final recommendations (blue arrows) to improve response are provided from the lowest temperature chain, and chosen such that they are not too close to each other and to experimental data (at least 20% difference). These recommendations are the “Recommendations for next cycle” depicted in Fig. 1. In this example, they represent protein expression levels that should be targeted to achieve predicted production levels. See Fig. 6 for an example of recommended protein profiles and their experimental tests.”

“Figure 1: **ART predicts the response from the input and provides recommendations for the next cycle.** ART uses experimental data (input and responses in the left side) to i) build a probabilistic predictive model that predicts response (e.g. production) from input variables (e.g. proteomics), and ii) uses this model to provide a set of recommended inputs for the next experiment (new input) that will help reach the desired goal (increase response/production). The input phase space, in this case, is composed by all the possible combinations of protein expression in this case (or transcription levels, promoters, etc. for other cases). The predicted response for the recommended inputs is characterized as a full probability distribution, effectively quantifying uncertainty in predictions. Instances refer to each of the different examples of input and response used to train the algorithm (e.g. each of the different strains and/or conditions, that lead to different production levels because of different proteomics profiles). See Fig. 2 for details on the predictive model and Fig. 3 for details on the recommendation strategy.”

P16I304 The simulation section is not very informative. It seems as if the authors are generating results that follow what already could be expected based on the mathematical approach. Basically, this section seems to be for debugging purposes. If x are proteomics data, how the E, M, D functions should be interpreted? Is there any reason that justifies the F_D formula? It would be more reasonable simulating some actual biological process, for instance biosynthetic pathways through their kinetic equations and their stationary proteomics data to the learning processes through several DBTL cycles.

In this section we use synthetic data to explore patterns (e.g. effect of more DBTL cycles on performance) that cannot be studied otherwise because the required amounts of data are not available. The formulas for these functions are abstractions that are useful to demonstrate the use of ART in a field (synbio) where data is scarce and expensive to generate. They just represent different levels of difficulty that can potentially be found. There is no systematic or standardized way to generate F_E , F_M or F_D from mechanistic principles or from kinetic equations. Deriving these functions from kinetic equations is, in itself, a very interesting and involved research project that is beyond the scope of this paper. We thank the reviewer for this interesting suggestion for our future research.

P19I34 The introductory paragraphs about limonene production are written like a review paper.

They should be summarized.

We have now summarized them. These paragraphs, however, are useful to putative readers with expertise in machine learning but not familiarized with synthetic biology. Hence we have kept most of the content so as to provide context and explain the importance of this project.

P20I378 The rationale of this section is not clear. We can assume that the second DBTL cycle was guided by the predictions obtained by the PCAP algorithm. Therefore, the fact that the proposed ART algorithm proposes the same recommendations does not show any clear improvement.

The reviewer is indeed right in pointing out that the second DBTL cycle for this project was guided by the PCAP algorithm, in the original study (Alonso-Gutierrez *et al*, 2015). We are using this data to test ART and check if it can recapitulate previous successes. It not only does this, but it presents three improvements with respect to PCAP:

1. It provides a quantitative prediction of the expected production in all of the input phase space, rather than qualitative recommendations as PCAP does.
2. ART is systematic and automated, requiring no human intervention to provide recommendations.
3. ART provides uncertainty quantification for the predictions, which PCAP did not do.

We have further clarified these points in the manuscript:

“ART is able to not only recapitulate the successful predictions obtained by PCAP improving limonene production, but also improves significantly on this method. Firstly, ART provides a quantitative prediction of the expected production in all of the input phase space, rather than qualitative recommendations. Secondly, ART provides a systematic method that is automated, requiring no human intervention to provide recommendations. Thirdly, ART provides uncertainty quantification for the predictions, which PCAP does not.”

P20I387 A cross validation of $R^2 = 0.44$ is not very convincing.

A cross validation of $R^2=0.44$ is not perfect, but it is good enough to provide effective recommendations, as shown in Figure 6. Biological predictions are particularly complicated and, given the limited amount of data available for training (27 instances), we believe this is a good result. We would like to point out that these are cross-validated predictions, which are done for data the model has not seen before. In any case, this exemplifies the point that the reviewer wanted substantiated above: we do not require quantitatively perfect predictions to effectively guide next steps.

P23I419 For the DBTL cycles of the hoppy beer example, it is not clear whether the authors mean that ART could not be trained (bad outcome on the test set) with the data of the first

cycle, or conversely it was successfully trained and provided good recommendations. This issue is repeated at other parts of this section. For instance, p251452 “despite the limited predictive power afforded by the cycle 1 data, ART recommendations guide metabolic engineering effectively”. This sentence is too ambiguous.

We apologize for the lack of clarity on this point. We meant to say that, despite not very accurate predictive models, ART can still suggest to explore new parts of the phase space that are known to eventually guide to increases in production. This effective guiding in spite of imperfect models is a recurring finding in this paper. While this can only be fully proved with more than two DBTL cycles, the fact that ART recommends exploring parts of the proteomics phase space such that the final protein profiles lie between the first cycle data and these recommendations is a good indication that this is the case (see the last paragraph in this section and Fig. S7).

That ART can guide metabolic engineering effectively is further proved in the parallel paper (Zhang *et al* 2019, NCOMMS-19-42070) where ART recommendations were followed and production was ultimately improved by 105%.

For the first point, we have expanded the discussion in the “Brewing hoppy beer without hops by bioengineering yeast” section:

“For both of the Pale Ale and Torpedo cases, ART recommends exploring parts of the proteomics phase space that surround the final protein profiles, which were deemed close enough to the desired targets in the original publication. Recommendations from cycle 1 and initial data (green and red in Fig. S7) surround the final protein profiles obtained in cycle 2 (orange in Fig. S7). Finding the final protein target becomes, then, an interpolation problem, which is much easier to solve than an extrapolation one.” (last paragraph in this section)

and we have added a new paragraph in the “Conclusion” section:

“ART has also been used to guide metabolic engineering efforts to improve tryptophan productivity in yeast, as shown in the experimental counterpart of this publication⁴². In this project, genome-scale models were used to pinpoint which reactions needed optimization in order to improve tryptophan production. ART was then leveraged to choose which promoter combinations for the five chosen reactions would increase productivity. ART's recommendations resulted in a 105% increase in productivity with respect to the initial base strain. We would expect further increases if more DTBL cycles were to be applied beyond the initial two (see the “Using simulated data to test ART” section). This project showcases how ART can successfully guide bioengineering processes to increase productivity, a critical process metric⁸¹ for which few systematic optimization methods exist. Furthermore, this project also demonstrates a case in which genetic parts (promoters) are recommended, instead of proteomics profiles as we did in the current paper. This approach has the advantage that it fully bridges the Learn and Design phases of the DBTL cycle, but it has the disadvantage that it may not fully explore the protein phase space (e.g. in case all promoters available are weak for a given protein).” (new paragraph in conclusion)

P26I459 The conclusions of the third example (dodecanol production) are too speculative. If the algorithm was not able to learn because the problem is hard and there were not enough data available, I think that the authors should refrain from “blaming” biological mechanisms like cell membrane production, which might be linked or not to dodecanol production.

We apologize for the misunderstanding: we did not mean to “blame” biological mechanisms, but rather mention mechanistic insights that may explain the poor predictive power of ART in this case. We have rephrased this paragraph stressing the difference between the facts (lack of predictive power) and our hypotheses (tight metabolic connection as the reason for low predictive power):

“The first challenge involved the limited predictive power of machine learning for this case. This limitation is shown by ART’s completely compromised prediction accuracy (Fig. 8). We hypothesize the causes to be twofold: a small training set and a strong connection of the pathway to the rest of host metabolism.”

“The dodecanol pathway depends on fatty acid biosynthesis which is vital for cell survival (it produces the cell membrane), and has to be therefore tightly regulated.⁷⁶ We hypothesize that this characteristic makes it more difficult to learn its behavior by ART using only dodecanol synthesis pathway protein levels (instead of adding also proteins from other parts of host metabolism).”

P27I495 Another issue, from the point of view of this reviewer, is that machine learning is used in order to learn the relationship between protein levels and production. However, any recommendation on levels obtained through the algorithm needs to be matched with the appropriate engineering (RBS, etc.) and therefore there is no direct method. In fact the authors mentioned cases where it was not possible to obtain the prescribed fold-increase in the protein expression.

The reviewer is right that, in this paper, ART has been used to provide a mapping between protein levels and production, and that obtaining the recommended protein levels is not trivial. Indeed, the third example (“Improving dodecanol production”), is used to show a case in which this problem arises. We believe it is important to present both the weaknesses and strengths of the method.

However, ART can also be used to recommend directly the pathway parts to be assembled (e.g. promoters), as has been demonstrated in Zhang *et al* (parallel experimental paper NCOMMS-19-42070). This approach has the advantage that it fully bridges the Learn and Design phases of the DBTL cycle, but it has the disadvantage that it may not fully explore the protein phase space (e.g. in case all promoters available are weak for a given protein).

We have collected these considerations in a new paragraph:

“Furthermore, this project also demonstrates a case in which genetic parts (promoters) are recommended, instead of proteomics profiles as we did in the current paper. This approach has the advantage that it fully bridges the Learn and Design phases of the DBTL cycle, but it has the disadvantage that it may not fully explore the protein phase space (e.g. in case all promoters available are weak for a given protein).”

P32 The limonene example (html) generated by a notebook cannot be visualized directly on GitHub. It would be better to store it as ipnb notebook.

We thank the reviewer for the suggestion. In addition to the html file, which cannot be visualized directly but only through downloading, we uploaded an .ipynb file to the GitHub repository.

P32 I was unable to access the hopless beer data link.

Our apologies, we should have specified that you need to get an account in the public ABF version of EDD (public-edd.agilebiofoundry.org). Accounts are free, but need to be created. The data should then be accessible. In any case, we have provided the data sets in a zipped file.

P32 Password-protected GitHub page does not seem to work.

We apologize for this inconvenience. Github changed its policy by requiring confirmation through the email associated to the anonymous reviewer accounts we created. We have added a zipped version of the files to this submission.

Reviewer 2:

In their manuscript Radivojević et al. present ART: A machine learning Automated Recommendation Tool for Synthetic Biology. ART combines machine learning and probabilistic modelling techniques to guide synthetic biology projects towards their specific engineering goal. ART learns from a first set of experimental data and recommends new strains to be engineered over several DBTL cycles.

The manuscript provides a solution (or the beginning of a solution) for a very important current challenge in Synthetic Biology: Namely how we can improve the learning step within the typical DBTL cycles that are required in metabolic engineering and Synthetic Biology, such that optimally performing strains can be reached faster and in a more systematic fashion.

In the short-term this would indeed readily allow to reduce cost for the successful implementation of new Synthetic Biology projects and eventually – in the long-term – enable fully automated metabolic engineering/Synthetic Biology labs.

The authors test ART with three simulated engineering efforts of different difficulty and three real ME examples from literature. While ART readily gives satisfactory suggestions after a few DBTL cycles for “easy” ME problems, it requires significantly more DBTL cycles for more difficult ones. Still, even in difficult cases ART could guide decision making and incremental strain improvements, where – without machine learning help – a project would likely be abandoned after a few cycles. Most importantly ART shows that difficult ME projects can be realised step-by step, but only if the metabolic engineer is prepared to undergo >10 reengineering cycles. This is essential knowledge for decision making in difficult but potentially impactful projects and challenges current ME practice (only going through very few DBTL cycles).

As such I consider the developed ART tool and the presented results a major improvement over state-of-the-art metabolic engineering practice.

Still, from an experimentalists point of view the authors leave several points open or under-discussed which reduces the potential usefulness of this manuscript for a user that wants to incorporate ART into their next Synthetic Biology project. But these points can be addressed.

We thank the reviewer for the positive comments. In particular the appreciation that we are addressing (and providing the beginning of a solution for) an important problem in synthetic biology: improving the Learn step in the Design-Build-Test-Learn cycle. We are also very thankful for the reviewer’s appreciation that this manuscript represents a “major improvement over state-of-the-art metabolic engineering practice”, and that it contains “essential knowledge for decision making”.

Major points

1. Experimental design of the starting (training) data: The manuscript does not examine or elaborate on the importance of properly designing the first set of input data. Given machine learning can provide predictive power by learning the underlying patterns in experimental data, I would assume that it is highly important that these data are as informative as possible from beginning on. E.g. starting data should sample as effectively as possible the parameter space to give the best impression of the underlying landscape.

As ART will eventually be used to guide new Synthetic Biology efforts, the experimentalist has some freedom in designing the starting data and needs advice on how to best do that (especially given that usually only low number of instances can be generated).

For example, in the original papers that were used in the current manuscript, the authors apply some level of experimental design (promoter strength, induction levels), but it is unclear if those considerations are actually optimal or useful for ART.

Denby et. al: “An initial set of 18 strains containing promoters predicted to span a wide range of expression strengths were constructed”

Alonso-Gutierrez et al.: “the enzymes in different gene clusters was generated through variation of promoter strength, different plasmid copy numbers, and under different induction timings and levels”

I suggest:

- The importance of effective experimental design and landscape sampling should be examined or illustrated by using different experimental designs in the simulated data case. e.g. testing the relation between “good” starting data and DBTL cycles needed.
- Further the experimental design should be discussed in the manuscript more thoroughly in form of guidelines or “things to consider” for experimentalists.

We agree with the reviewer in stressing the importance of the initial training data. We apologize that the following may not have been clear in our initial manuscript: we have a systematic method to produce starting data that leverages Design of Experiments expertise, in the form of Latin Hypercube drawing. Latin Hypercube drawing involves dividing the range of variables in each dimension into equally probable intervals and then choosing samples such that there is only one instance in each hyperplane (hyper-row/hyper-column) defined by those intervals. We have shown the importance of properly choosing the initial training dataset by comparing the results of using the systematic Latin Hypercube method with a naive approach that clusters all initial data points. We have provided more details regarding the Latin Hypercube method and created a new supplementary figure (Fig. S3) that shows that the systematic approach greatly outperforms the naive approach:

“We simulated the DBTL processes by starting with a training set given by 16 instances (see Fig. 1), obtained from Table 1 functions. Different instances, in this case, may represent different engineered strains or different induction or fermentation conditions for a particular strain. The choice of initial training set is very important (Supp. Fig. S3). The initial input values were chosen as Latin Hypercube⁶⁵ draws, which involves dividing the range of variables in each dimension into equally probable intervals and then choosing samples such that there is only one in each hyperplane (hyper-row/hyper-column) defined by those intervals. This ensures that the set of samples is representative of the variability of the input phase space. A less careful choice of initial training data can significantly hinder learning and production improvement (Supp. Fig. S3). In this regard, a list of factors to consider when designing an experiment can be found in the “Designing optimal experiments for machine learning” section in the supplementary material.”

and emphasized these points in the conclusion:

Also, it is highly recommendable to invest time in part characterization, pathway modularization and experimental design to fully maximize the effectiveness of ART, and data-driven approaches in general (see the “Designing optimal experiments for machine learning” section in the supplementary material for more details).

Furthermore, we have created a new section in the supplementary material entitled “Designing optimal experiments for machine learning” where we thoroughly stress the importance of metabolic engineers doing careful experiment design:

“Sample the initial phase space as widely as possible. Make sure you cover wide ranges for both input and response variables. Include bad (e.g. low production) and intermediate results as well as good ones (e.g. high production). This is the only way that the algorithms can learn to distinguish the inputs needed to reach any of these regimes. The Latin Hypercube¹⁴ that we used for the synthetic data case (“Using simulated data to test ART” section) is a good choice, but by no means the only one.”

2. The format of the final recommendation: The manuscript does not illustrate how the eventual output data (strain recommendations) look like; but the format of this recommendation has practical implications for the a priori pathway design (e.g. in terms of modularity). I assume the output is a list of strains linked to a list of different relative concentrations of the pathway’s components (e.g. enzymes). Are the new recommended relative concentrations set into the context of one or all (or none) of the previously existing strains?

I think a clear illustration (at least a supplementary figure) of the output data would help a potential biology user understand what type of suggestions ART gives and what type of fine-tuning is expected from a pathway. That would facilitate effective pathway design.

For example, the pathway needs to be sufficiently modular and the parts within it sufficiently characterised. As such, the usage and extension of e.g. MoClo toolboxes would be advisable.

We wholeheartedly agree with the reviewer’s suggestion to include an example of the output data, which can now be found as Supp. Fig. S5. It is important to note that the recommendations (output) come in the same form as the input: i.e. proteins if proteins are used as input as in the limonene case (“Improving the production of renewable biofuel” section), promoters if promoters as used as input as in the tryptophan case (parallel experimental paper by Zhang *et al*, NCOMMS-19-42070). However, we believe that providing the recommendations in the form of DNA parts (e.g. from the MoClo toolkit) must be tailored to each project specifically, and we cannot provide a general solution at this time. It is, however, an interesting challenge that we will consider in future work.

The manuscript mentions issues with ineffective a priori design, missing modularity and

unpredictable part behaviour (lines 497, 500 and 512) but these points should be focused and discussed in a separate section, ideally together with the experimental design (point 1). In that context it would be interesting to discuss how the use of machine learning in SynBio uncovers new (or let's say old but under-addressed) bottlenecks, like the need for well characterised parts/predictable part behaviour, insulated pathways, standardisations etc.

We have expanded our discussion on modularity, part characterization, and a priori design in the main text:

“These areas need further investment in order to accelerate bioengineering and make it more reliable, hence enabling design to a desired specification. Also, it is highly recommendable to invest time in part characterization, pathway modularization and experimental design to fully maximize the effectiveness of ART, and data-driven approaches in general (see the “Designing optimal experiments for machine learning” section in the supplementary material for more details).”

Furthermore, since the space in the manuscript is limited and we believe this is an important topic, we have created a new section in the supplementary material (“Designing optimal experiments for machine learning”) where we unify the considerations regarding experimental design, modularity, and part characterization, as suggested by the reviewer.

Minor points:

1. Definition of the term instances: while it becomes clearer towards the end of the manuscript (instances = refereeing to differently engineered strains and/or induction conditions thereof, that lead to different production readouts because of different expression level profiles; eventually used as a training set), the term instances should be defined when it is used for the first time e.g. in line 71 in the introduction or at least in line 310. In line 310 the sentence “with a training set given by 16 strains” lends itself to explain that those are referred to as instances.

In the conclusion section (line 517) the term instances seems to be replaced or reworded into “a set of vectors of measurements”.

In summary, the terminology for the input data needs to be clarified and described more consistently in order for the biology community to get a clear picture what data are fed into ART and accordingly which data need to be experimentally produced to start the ART-assisted DBTL.

We thank the reviewer for noticing this oversight: the term instance was never explicitly defined. We have now changed Fig. 1 to explicitly depict and name instances. Its caption has been changed to include a definition of the term “instance”:

“Figure 1: **ART predicts the response from the input and provides recommendations for the next cycle.** ART uses experimental data (input and responses in the left side) to i) build a probabilistic predictive model that predicts response (e.g. production) from input variables (e.g.

proteomics), and ii) uses this model to provide a set of recommended inputs for the next experiment (new input) that will help reach the desired goal (e.g. increase response/production). The input phase space, in this case, is composed by all the possible combinations of protein expression levels (or transcription levels, promoters, etc. for other cases). The predicted response for the recommended inputs is characterized as a full probability distribution, effectively quantifying prediction uncertainty. Instances refer to each of the different examples of input and response used to train the algorithm (e.g. each of the different strains and/or conditions, that lead to different production levels because of different proteomics profiles). See Fig. 2 for details on the predictive model and Fig. 3 for details on the recommendation strategy. An example of the output can be found in Supp. Fig. S5.”

2. More careful usage of the term -omics data: The manuscript uses the term -omics data when describing the input data that are required for ART. I think the term -omics data is a bit misleading (in disfavour of the manuscript). While not being wrong, it gives the impression that full -omics profiles of engineered strains are required for ART, which would be highly costly. I would clarify that and use “gene expression data” or “targeted proteomic data” for the pathway components of interest.

This is a very good point. We thank the reviewer for the suggestion and have clarified the term -omics data in Introduction:

“In the synthetic case and the three experimental cases where previous data is leveraged, we mapped one type of --omics data (targeted proteomics in particular) to production.”

and Conclusion:

“We have also explored several ways in which the current approach (mapping proteomics data to production) can fail when the underlying assumptions break down.”

3. All figures relating to the use of ART for real ME problems (5, 7 and 8) use the same graph type to illustrate the data: Observations are plotted against cross-validated predictions. One of the figure legends should be used to effectively explain this graph type to a biology user, how to interpret the data and why it is the best way to illustrate the data.

We have explained the graph type of showing observations against cross-validated prediction in the caption of Fig. 5:

“The cross-validation graphs (present in Figs. 7, 8, S8, S9 too) represent an effective way of visualizing prediction accuracy for data the algorithm has not yet seen. The closer the points

are to the diagonal line (predictions matching observations) the more accurate the model. The training data is randomly subsampled into partitions, each of which is used to validate the model trained with the rest of the data. The black points and the violins represent the mean values and the uncertainty in predictions, respectively.”

4. In figures 5,7 and 8 labelling of panels with a,b,c etc. instead of “top right” etc. would enhance readability of the figures.

We have now included the suggested labels into figures 5, 7, and 8.

5. Figure 6, the units for limonene production are missing.

We have added the units for limonene production in the revised manuscript.

Reviewers' Comments:

Reviewer #1:

Remarks to the Author:

The revised manuscript from Radivojevic et al greatly improves the description of ART, a machine-learning recommendation tool for metabolic engineering. As shown in the comments from the reviewers, there were in the original manuscript a substantial amount of points requiring clarification both in terminology and concepts. The authors have made a good job on addressing the comments and I think that the manuscript presents now their ART approach in a way that would be more insightful and relevant for a wider audience. Reading the rebuttal letter, it seems that the authors tend to think that some parts of the manuscript should be written in a way that caters to machine-learning specialists and others to experimentalists. I think that the manuscript is at its best when it refrains from this dualistic assumption and sticks to synthetic biology conventions as a common domain that fits better to the manuscript.

The authors provide a new piece of information that proves to be crucial in order to validate the approach. However, the way the authors are referring to this fifth experiment for tryptophan productivity can be misleading as it is referred as a second "parallel" publication (experimental "counterpart"). I am not particularly against such way of presenting the results, as the tryptophan might deserve a separate article on its own, but the authors would need to coordinate the efforts so that both publications can appear on time.

The addition of the "Designing optimal experiments for machine learning" expert advice section into Supplementary Materials, even if basic, provides useful protocols-like information.

Reviewer #2:

Remarks to the Author:

The authors addressed all comments properly and extended the manuscript accordingly, which makes ART easier to follow and eventually be applied by a biology user. In my opinion the manuscript is ready for publication.

REVIEWERS' COMMENTS:

Reviewer #1 (Remarks to the Author):

The revised manuscript from Radivojevic et al greatly improves the description of ART, a machine-learning recommendation tool for metabolic engineering. As shown in the comments from the reviewers, there were in the original manuscript a substantial amount of points requiring clarification both in terminology and concepts. The authors have made a good job on addressing the comments and I think that the manuscript presents now their ART approach in a way that would be more insightful and relevant for a wider audience.

We thank the reviewer for the positive comments.

Reading the rebuttal letter, it seems that the authors tend to think that some parts of the manuscript should be written in a way that caters to machine-learning specialists and others to experimentalists. I think that the manuscript is at its best when it refrains from this dualistic assumption and sticks to synthetic biology conventions as a common domain that fits better to the manuscript.

We believe that it is in the intersection of synthetic biology and machine learning that great scientific opportunities lie (as we have discussed in Carbonell et al, ACS Synth. Biol. 2019). However, there are few experts in both areas. We hope this paper can bridge both disciplines, encouraging their interaction further.

It is for this reason that we have made a significant effort to make each section of the paper accessible to *both* machine-learning specialists and synthetic biologists. We hope this paper can be a gateway for synthetic biologists willing to learn about machine learning, and vice versa. We think that limiting it to one type of audience would detract from its scientific value.

We have, however, made several further changes to facilitate comprehension for synthetic biologists (in blue in the updated version).

The authors provide a new piece of information that proves to be crucial in order to validate the approach. However, the way the authors are referring to this fifth experiment for tryptophan productivity can be misleading as it is referred to as a second “parallel” publication (experimental “counterpart”). I am not particularly against such a way of presenting the results, as the tryptophan might deserve a separate article on its own, but the authors would need to coordinate the efforts so that both publications can appear on time.

We have coordinated with the Nature Communications editors to enable the simultaneous publication of both papers.

The addition of the “Designing optimal experiments for machine learning” expert advice section into Supplementary Materials, even if basic, provides useful protocols-like information.

Thanks for the positive assessment: this section came about because of comments by reviewer #2, and we agree that it provides useful information.

Reviewer #2 (Remarks to the Author):

The authors addressed all comments properly and extended the manuscript accordingly, which makes ART easier to follow and eventually be applied by a biology user. In my opinion the manuscript is ready for publication.

We thank the reviewer for the kind words.